# Multi-Task Consistency-based Detection of Adversarial Attacks

## Abstract

Deep Neural Networks (DNNs) have found successful deployment in numerous vision perception systems. However, their susceptibility to adversarial attacks has prompted concerns regarding their practical applications, specifically in the context of autonomous driving. Existing research on defenses often suffers from cost inefficiency, rendering their deployment impractical for resource-constrained applications. In this work, we propose an efficient and effective adversarial attack detection scheme leveraging the multi-task perception within a complex vision system. Adversarial perturbations are detected by the inconsistencies between the inference outputs of multiple vision tasks, e.g., objection detection and instance segmentation. To this end, we developed a consistency score metric to measure the inconsistency between vision tasks. Next, we designed an approach to select the best model pairs for detecting this inconsistency effectively. Finally, we evaluated our defense by implementing PGD attacks across multiple vision models on the BDD100k validation dataset. The experimental results demonstrated that our defense achieved a ROC-AUC performance of 99.9% detection within the considered attacker model.

## 1 Introduction

The camera-based perception system is critical to enable automated driving (AD). Indeed, camera is the only sensor able to read traffic signs, identify lane markings or drivable areas, and see traffic light colors. To perform such perception tasks (e.g., object detection, classification, segmentation), a wide range of machine learning models were developed, each with its own objective and network architecture (Zou et al., 2023). For example, from an input image, 2D object detection models output bounding boxes, while semantic segmentation models output masks, or multi-object tracking models output track identifiers. The model outputs help to understand the scene and allow the automated vehicle to maneuver appropriately.

However, camera inputs can be maliciously manipulated to affect the performance of perception tasks, or even downstream tasks of automated vehicles (e.g., path planning, motion control). The idea of adversarial inputs (commonly called *adversarial examples*) is to add specially-crafted noise to images such that the underlying machine learning models do not perform as originally intended (Madry et al., 2018). Adversarial examples have been demonstrated in the form of full image perturbations or patches, realized digitally or physically, and with some high attack success rate and universality (Chow et al., 2020). Because of their low level of sophistication and effectiveness, it is key to deploy defenses to protect automated vehicles against such threats. Defenses range from preemptive techniques (e.g., adversarial training (Shafahi et al., 2019), certified robustness (Xiang et al., 2024)) to reactive techniques (e.g., real-time detection of perturbations (Xiang et al., 2022), image compression (Das et al., 2018)).

In this paper, we focus on reactive techniques, aiming at real-time detection of perturbations, because it does not require any adversarial data generation or additional training. Especially, we propose to leverage the output of multiple perception tasks to identify perturbations on every image prior to use by downstream tasks. Prior work showed the effectiveness of checking inconsistencies of edge extractions between outputs of semantic segmentation and depth estimation (Klingner et al., 2022), but with some limitations. Their inconsistency check only detects adversarial perturbations on the entire image and might show limited performance on local perturbations. Therefore, we propose

a consistency-based detection technique that is effective regardless of the perturbation's location. As long as the perturbation causes inconsistent inference output across models, locally or globally, our defense can capture the inconsistency. Especially, we demonstrate the benefits of cross-model consistency by using 2D object detection and instance segmentation models. Indeed, 2D object detection models are commonly used in AD to detect obstacles or traffic signs, and then to convert 2D bounding boxes to 3D bounding boxes (Feng et al., 2020; Arnold et al., 2019). Instance segmentation is also used in AD to provide finer object boundaries (Zhou et al., 2020). Both model share the objective of detecting objects, and hence, can be used to identify inconsistencies.

Our **contributions** are as follows:

- We propose a lightweight consistency detector based on outputs from object detection and instance segmentation models.
- We develop a technique to select the optimal model pair, deriving requirements w.r.t model architecture.
- We define a metric to capture the consistency score between two models' output.
- We generate and publish an adversarial BDD100k dataset to assess the effectiveness of our defense, and allow reproducibility and comparison of future defenses.

## 2 SYSTEM MODEL

### 2.1 VISION MULTI-TASK SYSTEM

Perception systems perform multiple vision tasks such as object detection, segmentation, and depth estimation. Because of its better generalization performance and efficiency (Guo et al., 2020b), one architecture considered for automated driving is Multi-Task Learning (MTL) (Miraliev et al., 2023). A common approach in MTL is to have a shared feature extractor and multiple task-specific heads (Caruana, 1997; Kokkinos, 2017; Lu et al., 2017). In this paper, because our detection method must work with MTL and non-MTL architecture, our architecture consists of one model per task. With this flexible approach, we can evaluate the performance of our detector when the tasks share (or not) the same backbone. Indeed, the attack success rate strongly correlates with the architecture similarity between tasks as highlighted by Xie et al. (2017). Interestingly, from a security perspective, it may be more robust to have an architecture with different backbone per task than a common backbone architecture for all tasks (like in the MTL architecture).

### 2.2 ATTACKER MODEL

We follow the same attacker model as defined by Xiang et al. (2022), where the attacker performs a white-box attack (i.e., has access to the model's architecture and weights). We assume a model $\mathbb{F}$ with an underlying data distribution $\mathcal{D}$ over pairs consisting of image $\mathbf{x}$ and its corresponding ground truth $y$. $\mathcal{X}$ denotes the image space. The attacker adds the perturbation $\delta$ to the genuine image $\mathbf{x}$ to create an adversarial image ($\mathbf{x}' = \mathbf{x} + \delta$) (with $||\delta||_p \leq \epsilon$, where $\epsilon$ is the bound on the $L_p$ norm perturbation) such as $\mathbf{x}' \in \mathcal{A}(\mathbf{x}) \subset \mathcal{X}$, where constraint $\mathcal{A}$ defines the attacker's capability. The goal of the attacker is to minimize the alteration of the genuine image $\mathbf{x}$ while ensuring the attack succeed, and is formulated as:

$$\min ||\mathbf{x}' - \mathbf{x}|| \; s.t. \; \mathbb{F}(\mathbf{x}') \neq \mathbb{F}(\mathbf{x}) \tag{1}$$

where, $\mathbb{F}(\mathbf{x}') \neq \mathbb{F}(\mathbf{x})$ can be the removal or injection of bounding boxes/masks.

To achieve her goal, the attacker uses a projected gradient descent (PGD) attack (Madry et al., 2017).

$$\mathbf{x}'_{t+1} = \Pi_{\mathbf{x}+\mathcal{X}}(\mathbf{x}'_t + \alpha sgn(\nabla_{\mathbf{x}} L(\theta, \mathbf{x}', y))) \tag{2}$$

where, $L(\theta, \mathbf{x}', y)$ is the global loss function defined as the sum of classification loss and localization loss ($L = L_{cls} + L_{loc}$). Hence, the perturbation targets a misclassification or mislocalization.

## 3 MULTI-TASK CONSISTENCY

We first define the multi-task consistency score between model outputs across different vision tasks. In particular, we use object detection (OD) and instance segmentation (SEG) as example vision tasks in this paper. Then, we explain how to use the consistency score to detect adversarial perturbations.

**Object Detection**    **Instance Segmentation**

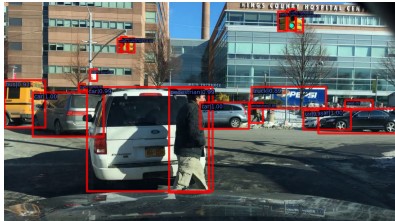    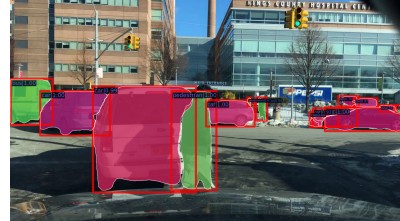

(a) Clean image    (b) Clean image

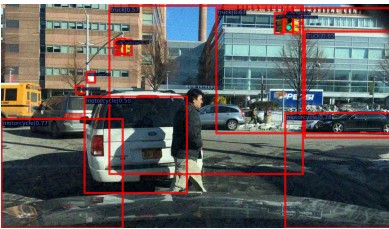    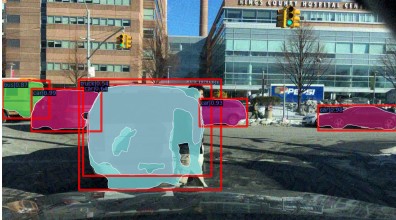

(c) Perturbation optimized for object de-    (d) Perturbation optimized for object de-
tection    tection

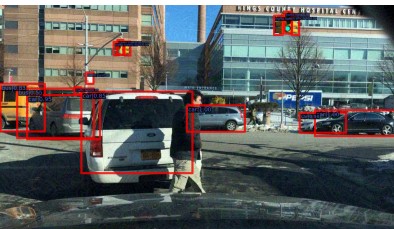    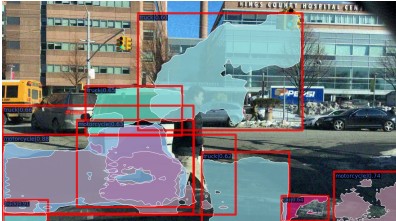

(e) Perturbation optimized for instance    (f) Perturbation optimized for instance seg-
segmentation    mentation

Figure 1: Impact of adversarial perturbation on the vision models

## 3.1 CONSISTENCY BETWEEN VISION TASKS

As shown in Figure 1, the inference outputs for object detection and instance segmentation on clean images exhibit overall consistency. Indeed, the object bounding boxes match with the object masks. However, on the perturbed images, discrepancies arise. For instance, in Figures 1c-1d, the perturbation optimized for the object detection model successfully deceived the object detector, leading to numerous false positive and false negative predictions. On the other hand, the same perturbation did not fool the instance segmentation model, which accurately predicted the bounding boxes and masks[1]. Similar impact is observed in Figure 1e-1f where the perturbation is optimized for instance segmentation. In fact, we can identify two types of consistency between the model outputs:

- Location Consistency: refers to detecting an object at the same location within an input image using both an object detection model and an instance segmentation model. It involves calculating the Intersection over Union (IoU) between each detected object from both models. If the IoU exceeds a predefined threshold (e.g., 50%), the object pair is considered *location consistent*.
- Semantic Consistency: goes beyond location and ensures that the labels of the object pair are identical as well. In this paper, we consider a detection as *consistent* if both location and semantic consistency are proven.

**Consistency Score.**    In this work, we call *consistent detection* ($CD$) a matching pair of box and mask (location and label wise). In order to measure the overall consistency of a single image, Equation 3 defines the *Consistency Score* $C_{task}$ as the ratio of total number of consistent detection over the total number of detection from either model ($N_{task}$).

---

[1]We note a slight impact on the foreground objects' masks.

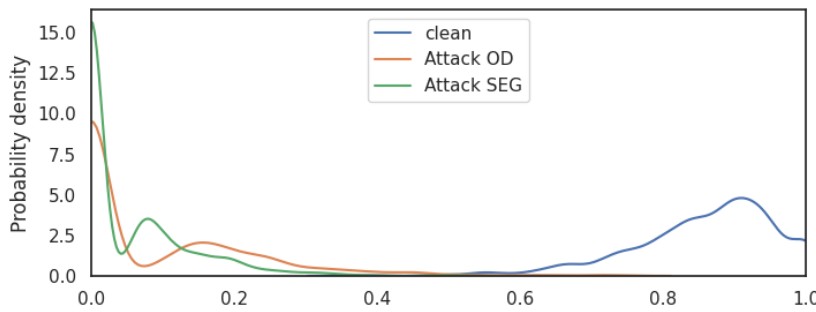

Figure 2: Empirical study of the consistency score distribution for (FRCNN R50, MRCNN R50) pair on BDD100k dataset. Blue line shows consistency scores for clean images. Orange line shows consistency scores for perturbed images (target OD). Green line shows consistency scores for perturbed images (target SEG). The clear divergence between distributions, confirms the ability of our detector to identify perturbations.

$$C_{\text{task}} = \frac{|CD|}{N_{\text{task}}} \quad \text{task} \in \{det, seg\} \tag{3}$$

Then, as in Equation 4, we define **consistency score** $C$ as a harmonic mean of $C_{\text{det}}$ and $C_{\text{seg}}$ to measure the overall consistency of the inferences on input images by both models.

$$C = \frac{2 \cdot C_{\text{det}} \cdot C_{\text{seg}}}{C_{\text{det}} + C_{\text{seg}}} \tag{4}$$

**Empirical Study.** We performed an empirical study on the consistency score distribution for a clean image dataset and perturbed image datasets. The clean dataset consists of 1000 images from BDD100k validation dataset. The perturbed dataset is created by applying the PGD attack on each image in the clean dataset. Then, we calculated the consistency score of (FRCNN R50, MRCNN R50) model pair using Equation 4 on each image of the datasets and plot the distribution using kernel density estimation. From Figure 2, we observe that the model pair on clean images have higher consistency score, while perturbed images have much lower consistency score. This implies that we can distinguish between clean and perturbed images using the consistency score. We present other models distribution plots in Appendix A.4.

## 3.2 CONSISTENCY SCORE BASED ATTACK DETECTION

Inspired by the above observation, we propose a consistency score based adversarial attack detection scheme illustrated in Figure 3.

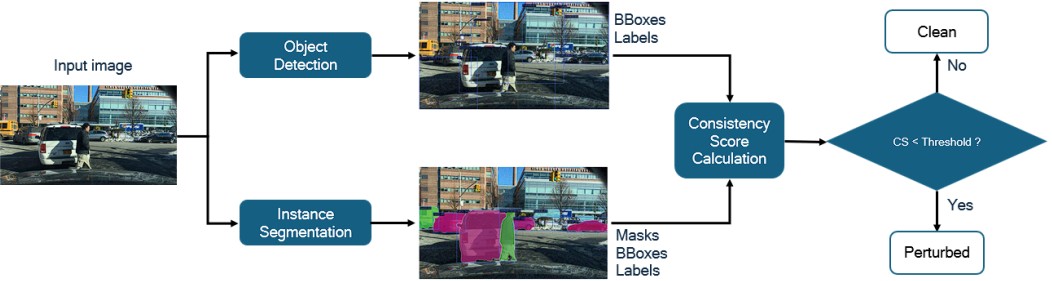

Figure 3: Pipeline of consistency score based adversarial perturbation detection

**Notations.** In order to formulate the problem, we denote the output of the object detection model as a set of annotations of detected objects $S_{\text{det}} = \{(\text{BBox}_{\text{det},i}, \text{Label}_{\text{det},i}) | i = 1, ... K_{\text{det}}\}$ where $\text{BBox}_{\text{det},i}$ is the bounding box coordinates for the $i$-th detection, $\text{Label}_{\text{det},i}$ is its corresponding class label, and $K_{\text{det}}$ is the total number of detection by the object detection model. Similarly, we denote the output of the instance segmentation model as $S_{\text{seg}} = \{(\text{BBox}_{\text{seg},j}, \text{Label}_{\text{seg},j}) | j = 1, ... K_{\text{seg}}\}$.

**Step 1: Consistency Score Calculation.** Following Equation 3, the consistency score is calculated between the two tasks output. In Appendix A.1, we propose Algorithm 1 as an implementation of the Consistency Score Calculation module of Figure 3.

**Step 2: Adversarial determination.** With the consistency score generated for the input image, the next step is to decide if it is a clean or perturbed image. As shown in Figure 3, a threshold-based binary classification takes the consistency score as input. If the consistency score is lower than the predefined threshold, the input is labelled as "perturbed". As shown in Figure 2, setting a high cut-off threshold (e.g., 0.75) would trigger false positives. Conversely, selecting a low threshold (e.g., 0.2) would trigger false negatives. Therefore, there is a trade-off between false positive rate and false negative rate. Implementers would have to pick the appropriate threshold using known techniques (Lan et al., 2020).

**Cross-task model selection.** When designing a multi-task consistency detector, it is important to select the appropriate models used for each task. Indeed, the two models[2] could share the same backbone and underlying structure, or only share the same backbone, or share similar backbone but with different layer depth. We aim at answering the question "What model architectures or parameters affect the ability to detect adversarial inputs via multi-task consistency?". For example, should the feature extractors be different? if so, to what extent? Ghamizi et al. (2022) hinted that one should carefully select the auxiliary tasks added to reduce model vulnerability. Indeed, the addition of auxiliary tasks can have negative effects (e.g., larger model size, slower convergence of the common encoder layers, deterioration of clean performance). They raised the (still open) question of how to select the combination that yields the lowest vulnerability. One could think that picking the most adversarially robust backbone would be preferable. For example, when investigating ResNet50 and ResNet101 backbones, the only difference is that ResNet101 has 23 conv4_x layers while ResNet50 has 6 (so a total of 51 additional convolution layers as the name indicates). This means that ResNet101 has larger receptive fields than ResNet50. As shown by Xiang et al. (2024), smaller receptive fields impose a bound on the number of features that can be corrupted, hence more adversarially robust. This could justify the use of ResNet50 backbone over ResNet101. However, in our context, we select models that, even if fooled by the attack, yield to inconsistent outputs. So, having two weak models could be acceptable as long as their outputs are inconsistent.

## 4 EXPERIMENTS

In this section, we outline the implementation details of the datasets, models, attack parameters, and evaluation metrics used to ensure reproducibility. We then present and discuss the experimental results of the multi-task consistency-based detector. Additionally, we offer recommendations for a cross-model strategy to select the best model pairs for the detector.

### 4.1 IMPLEMENTATION DETAILS

**Datasets.** Our evaluation relies on a set of genuine and adversarial datasets based on the BDD100k dataset. The description of the BDD100k dataset can be found in the Appendix( A.2).

**Models.** We use 11 existing models from BDD100k model zoo (Huang, 2021) and from mmdetection 2.0 framework (Chen et al., 2019): six models for object detection (OD) and five models for instance segmentation (SEG). All models are fine-tuned on the BDD100k dataset. Our selection of models aims to maximize the diversity of models for a given vision task to understand how it affects the performance of our defense. Indeed, an adversarial attack may transfer from one model to

---

[2]For the sake of conciseness, we restrict ourselves to a model pair. However, the system can be extended to larger tuple (see Section 5).

another if their architecture are similar. Therefore, we chose our models based on a set of criteria. The first one is the type of architecture (e.g., transformer or CNN). A second criteria is the depth of the backbone (ResNet50 versus ResNet101). The last criteria is to ensure a diversity of heads among the models (e.g., FRCNN versus RetinaNet).

**Attack.** We utilized 1,000 clean images from the BDD100k instance segmentation validation dataset for our attack. This dataset was selected due to its comprehensive annotations, which include both segmentation masks and bounding boxes, allowing us to fairly assess the impact on both object detection (OD) and segmentation (SEG) models. We then applied the PGD-40 attack (40 iterations with a perturbation strength $\epsilon = 16/255$) to each of the eleven models. This resulted in 11 adversarial datasets: six from attacking the OD models and five from attacking the SEG models. We use the clean dataset alongside these 11 adversarial datasets to evaluate the performance of the models and our detection scheme.

**Evaluation Metrics.** To evaluate the prediction performance of the models on both the clean dataset and the eleven adversarial datasets, we utilize the mean Average Precision (mAP), a widely accepted metric for assessing computer vision models. For evaluating our detection scheme, we employ the receiver operating characteristic (ROC) curve, a popular metric that illustrates the performance of a classification model across all classification thresholds. The area under the curve (AUC) provides a measure of our adversarial attack detection performance.

## 4.2 EXPERIMENTAL EVALUATION

First, we study the transferability of the attack. Next, we assess the performance of our detector scheme in detecting perturbations. We then discuss the cross-model strategy. Finally, we compare our detector with one state-of-the-art defense.

### 4.2.1 PREDICTION PERFORMANCE UNDER ATTACK

Table 1 shows the mAP for each model across twelve test datasets. The table's diagonal highlights that the attack is most effective on the target model for which the perturbation is optimized. For instance, the attack on the FRCNN R50 model decreases its mAP from 30.2 to 0.18. Comparable performance declines can also be noted for the other models under attack, which was expected given that our attack is a white-box PGD attack on the target models.

The perturbations demonstrate transferability across models with similar network architectures, regardless of the task. For instance, the adversarial dataset generated by attacking the OD model FRCNN R50 decreases the mAP of the SEG model MRCNN R50 from 19.8 to 1.5. Conversely, the attack on MRCNN R50 reduces the mAP of FRCNN R50 from 30.2 to 8.8. This indicates that the perturbation can transfer to different tasks or models with the same backbone architectures. Transferability is also observed in models that share the same backbone type but differ in depth. As shown in Table 1, attacks on models with an R50 backbone (see columns) can transfer to models with an R101 backbone (see rows), and vice versa. However, for models with the same baseline architecture but different backbones, such as FRCNN R50 and FRCNN SwinT, or RetinaNet R50

Table 1: Impact of the attack on the mAP of vision models

| Task | Vision Models | Clean mAP (↑) | Attack Object Detection (↓) | | | | | | Attack Segmentation (↓) | | | | |
|---|---|---|---|---|---|---|---|---|---|---|---|---|---|
| | | | F R50 | F R101 | F SwinT | R R50 | R R101 | R PVT | M R50 | M R101 | G R50 | G R101 | M2F SwinT |
| OD | F R50 | 30.2 | **0.18** | 5.7 | 18.4 | 0.34 | 3.6 | 11.8 | 8.8 | 9.2 | 8.8 | 10.0 | 24.7 |
| | F R101 | 30.3 | 7.5 | **0.17** | 18.5 | 3.2 | 1.0 | 13.0 | 14.5 | 6.4 | 14.1 | 8.06 | 25.0 |
| | F SwinT | 31.8 | 17.0 | 16.5 | **1.5** | 11.5 | 12.0 | 14.8 | 20.7 | 18.6 | 20.7 | 19.0 | 22.4 |
| | R R50 | 28.7 | 2.2 | 4.4 | 17.0 | **0.01** | 2.7 | 10.2 | 7.1 | 7.4 | 7.0 | 8.9 | 23.1 |
| | R R101 | 29.2 | 7.7 | 2.2 | 17.9 | 2.63 | **0.02** | 11.9 | 14.0 | 5.6 | 13.5 | 7.1 | 24.3 |
| | R PVT | 29.8 | 12.8 | 13.0 | 18.4 | 7.5 | 8.9 | **0.04** | 18.2 | 15.0 | 17.8 | 15.0 | 24.8 |
| SEG | M R50 | 19.8 | 1.5 | 2.6 | 10.1 | 0.6 | 2.6 | 7.1 | **0.01** | 1.6 | 0.5 | 2.2 | 13.4 |
| | M R101 | 20.5 | 4.2 | 1.8 | 10.3 | 2.5 | 1.4 | 8.0 | 4.4 | **0.01** | 4.2 | 0.72 | 13.4 |
| | G R50 | 20.1 | 1.7 | 3.1 | 10.7 | 0.62 | 2.9 | 6.9 | 0.73 | 1.8 | **0.01** | 2.1 | 13.3 |
| | G R101 | 20.7 | 4.2 | 2.0 | 10.3 | 2.6 | 1.7 | 7.6 | 4.4 | 0.3 | 4.1 | **0.01** | 13.2 |
| | M2F SwinT | 21.0 | 9.4 | 9.1 | 7.1 | 7.3 | 7.8 | 9.7 | 9.7 | 7.9 | 10.1 | 9.1 | **2.8** |

Acronyms: Object Detection (OD), Instance Segmentation (SEG), FRCNN (F), RetinaNet (RN), MRCNN (M), GCNET (G), Mask2Former (M2F)
**A bold value** is the lowest mAP score among all targeted models for a given adversarial dataset.
An underlined value indicates the adversarial dataset successfully dropped the mAP score of the targeted model below 5 mAP.

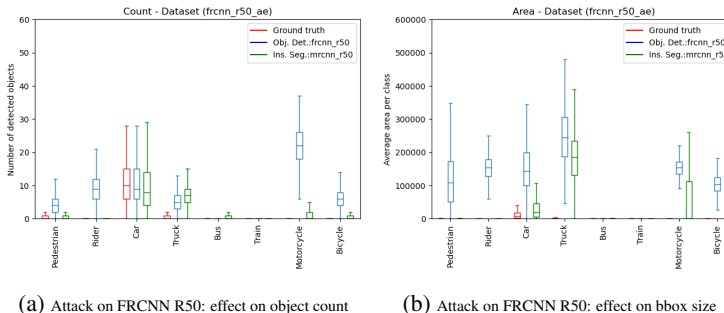

(a) Attack on FRCNN R50: effect on object count    (b) Attack on FRCNN R50: effect on bbox size

Figure 4: Perturbations optimized for FRCNN R50 transfer to MRCNN R50, impacting them differently in terms of number of detected objects and their sizes.

and RetinaNet PVT, the transferability is less evident. This indicates that the backbone plays a more crucial role in the transferability of the attack.

While perturbations can transfer between different models and tasks, their fine-grained impact varies significantly across models. This variation is evident in several aspects, such as the number of objects detected or the area of the bounding boxes. Figure 4 illustrates this using one model pairs: (FRCNN R50, MRCNN R50). Subfigure 4a shows the distribution of the number of objects for each category given the adversarial dataset optimized on FRCNN R50. The attack generates significantly more objects on FRCNN R50 than on MRCNN R50, especially for categories like rider and motorcycle. Subfigure 4b indicates that the attack also causes larger objects for FRCNN R50 in most categories, while for MRCNN R50, this effect is seen only in the truck and motorcycle categories. This demonstrates that even when perturbations transfer, they can lead to inconsistent impacts on different models[3].

### 4.2.2 PERTURBATION DETECTION PERFORMANCE

We evaluated the performance of our detector across 30 (6 OD × 5 SEG) model pairs. As previously noted in Figure 2, we aim for a model pair that exhibits a high consistency score (CS) on clean inputs while a lower score on adversarial inputs, facilitating the identification of perturbations. Figure 5 (top) illustrates the average CS of model pairs across the three datasets (clean, attack OD, attack SEG). The blue stars represent the average CS for clean inputs. Generally, the CS for clean inputs is high, especially for model pairs with similar baseline architectures (RCNN) and backbones (ResNet), which can extract consistent features from the clean inputs, resulting in consistent outputs. Model pairs with different architectures or backbones exhibit slightly lower CS due to their varying feature extraction capabilities, leading to inconsistent outputs. The red squares represent the CS for adversarial datasets optimized on OD models. As discussed in the previous section, similar architectures (RCNN and ResNet) result in high transferability but also high inconsistency, causing CS to drop as low as 0 for those model pairs. For model pairs with different architectures, the attack shows less transferability, and thus, higher consistency. Similar findings are observed when attacking SEG models (green circles). The full analysis can be found in Appendix A.3.

The AUC curves in Figure 5 (bottom) demonstrate that all model pairs achieve an AUC greater than 85%, with most exceeding 95%, when either model of the pairs is attacked. This highlights the exceptional performance of our consistency-based detector in identifying perturbations. Model pairs with similar backbone types (ResNet) and baseline architecture (RCNN) exhibit the highest performance, achieving an AUC of 99.9%. Again, it shows that, although the attack can easily transfer between these models, this transferability leads to distinct variations in the number, label, and size of the detected objects. These variations result in a higher level of inconsistency, which our detector can effectively identify. In contrast, model pairs with different backbones or baseline architectures exhibit low transferability and low inconsistency, resulting in a relatively lower AUC.

---

[3]See Appendix A.3 for full transferability analysis across all model pairs considered.

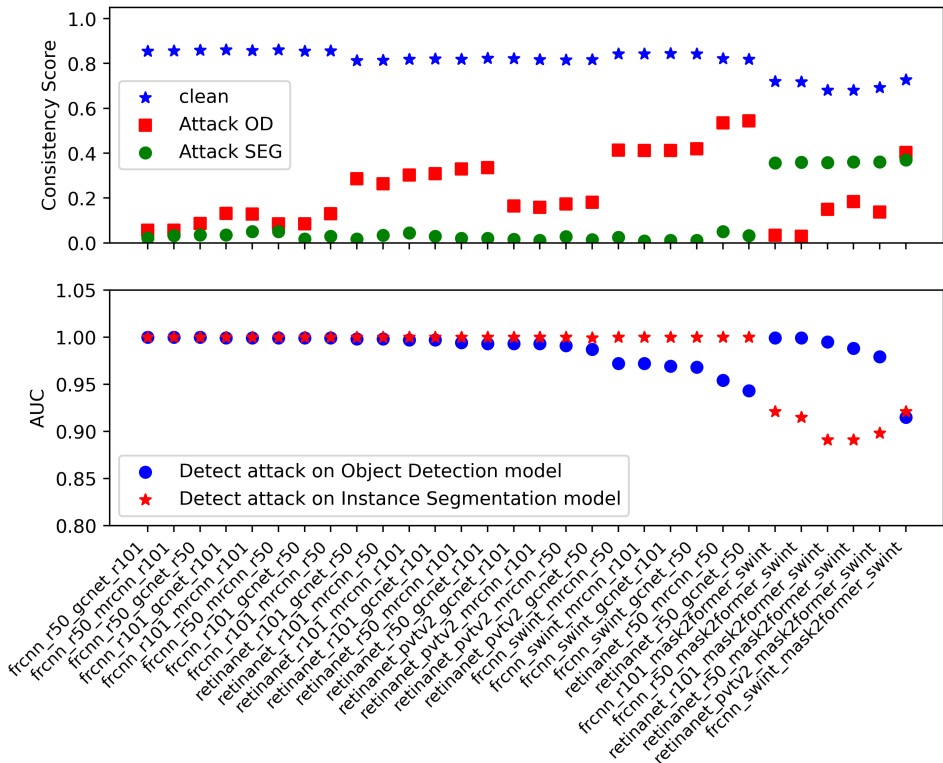

Figure 5: Top: Consistency Score for all model pairs. The lower the better the pair is for our detector. Bottom: The AUC for all model pairs. The higher the better the pair is our detector.

Table 2: Effect of perturbation strength $\epsilon$ on mAP. Target model is FRCNN R50.

| Model | Perturbation Strength | | | | | |
|---|---|---|---|---|---|---|
| | clean | 1 | 2 | 4 | 8 | 16 |
| **FRCNN R50** | 30.2 | 28.4 | 25.4 | 18.7 | 4.2 | 0.18 |
| **MRCNN R50** | 19.8 | 18.9 | 17.0 | 13.4 | 6.2 | 1.5 |

Next, we are interested to learn how the perturbation strength of the attack can impact the prediction performance of the models and the detection performance of our detector. We evaluate the robustness of the models against attack size $\epsilon \in \{1/255, 2/255, 4/255, 8/255, 16/255\}$. Here, we use one of the best model pair (FRCNN R50, MRCNN R50) as an example. More results of other model pairs can be found in the Appendix A.4. As shown in Table 2, the mAP of both models decreases as the perturbation strength increases, which is expected. Conversely, as illustrated in Figure 6b, the detection performance in terms of AUC increases. This is the desired behavior because stronger perturbation leads to greater inconsistency (lower consistency score as in Figure 6a) between the outputs of model pairs, resulting in higher detection performance for our detector.

**Takeaway on multi-task architecture.** The empirical analysis indicates that our detector performs optimally when model pairs exhibit high inconsistency in their outputs. Under our attacker model, the most effective model pairs are those with similar architectures and backbones, as they demonstrate high adversarial transferability but also high inconsistency.

### 4.2.3 COMPARISON TO OTHER DEFENSES

In this section, we compare our detector with the adversarial training method RobustDet, as proposed by Dong et al. (2022). For a fair comparison, we applied RobustDet to FRCNN R50 which resulted in a robust model named RobustFRCNN. More details about our implementation can be found in

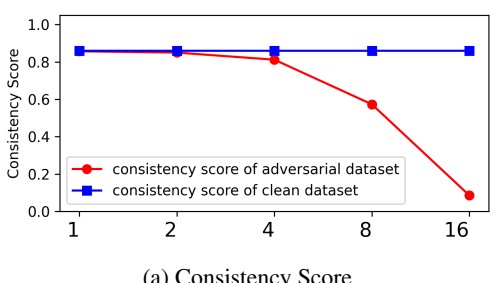 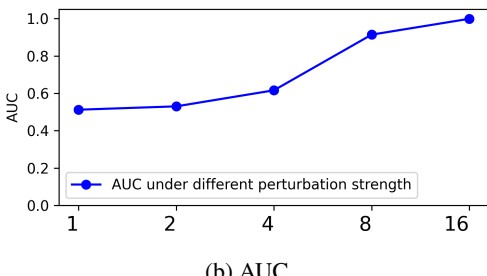

(a) Consistency Score

(b) AUC

Figure 6: Impact of perturbation strength $\epsilon$ on our detector (consistency score AUC)

Table 3: Impact of the attack on the mAP of regular and RobustFRCNN

| Model | Clean | | | | Attack | | | |
|---|---|---|---|---|---|---|---|---|
| | mAP | mAP$_{small}$ | mAP$_{medium}$ | mAP$_{large}$ | mAP | mAP$_{small}$ | mAP$_{medium}$ | mAP$_{large}$ |
| **FRCNN R50** | 30.2 | 12.4 | 34.6 | 54.4 | 0.18 | 0.07 | 0.24 | 0.33 |
| **RobustFRCNN** | 19.8 | 8.1 | 22.4 | 36.7 | 6.2 | 2.4 | 7.3 | 11.9 |

Appendix B.2.1. First, we evaluated the prediction performance of two models based on FRCNN R50: the standard model and RobustFRCNN. Table 3 presents the performance of both models under clean and adversarial datasets. For the standard model, we used the same adversarial dataset as previously mentioned. For the robust model, we applied PGD attack using same attack parameters. Table 3 shows that the standard model experiences a significant performance drop due to the attack, compared to the robust model. Specifically, its mAP decreases from 30.2 to 0.18, while for the RobustFRCNN, it decreases from 19.8 to 6.2. Hence, RobustDet technique enhances the model's adversarial robustness. It is worth noting that the clean mAP for the robust model is not high, indicating there is still room to adjust the training parameters to improve both its clean and adversarial performance.

Next, we compare our detector with RobustDet. Our detector functions as a binary classifier, determining whether an input is adversarial or not. In contrast, RobustDet, similar to adversarial training, enhances the model's robustness. To ensure a fair comparison, we introduce the metric *Detection Rate*, which represents the true positive rate for a given adversarial dataset. Specifically, we utilize one of our best model pairs (FRCNN R50, MRCNN R50) for our detector and assess its detection rate on the adversarial dataset for FRCNN R50. For RobustFRCNN, we evaluate its performance by calculating the consistency score between its output and the ground truth annotations under adversarial conditions. A high consistency score indicates that RobustFRCNN successfully mitigates the perturbation, whereas a low score signifies failure. Therefore, the detection rate is the ratio of adversarial inputs with a consistency score above the consistency threshold.

As shown in Table 4, our consistency-based detector successfully identifies all adversarial inputs (99.9%) in the adversarial datasets, thanks to the high inconsistency between the outputs of the model pair. In contrast, RobustFRCNN performs poorly (mAP = 6.2), failing to ensure prediction outputs align with the ground truth, resulting in a very low detection rate (19%). On top of being less able to detect adversarial inputs, RobustFRCNN employs a dynamic convolution kernel that is four times the size of a regular convolution kernel, significantly increasing its model size. In comparison, our detector has a combined weight size of only 350MB for OD and SEG. Finally, our detector achieves faster inference speeds on the same hardware due to its less complex architecture. In summary, our detector demonstrates stronger performance compared to RobustDet.

Table 4: Performance of RobustFRCNN vs our detector

| Defense | Detection Rate | Model Weight Size (MB) | Inference Speed (FPS) |
|---|---|---|---|
| **RobustFRCNN** | 19 | 643 | 11 |
| **Our detector** | **99.9** | **350** | **20** |

## 5 OPEN CHALLENGES

**Stronger attacker model.**   Our attacker model targets only one perception task. A stronger attacker could target both tasks. Prior work have demonstrated attacks fooling both semantic segmentation and object detection with some success (Xie et al., 2017). More generally, techniques were designed to improve the adversarial transferability cross-model or cross-task (Gu et al., 2023; Wei et al., 2024; Hu et al., 2024; Lu et al., 2020). We expect that an attacker that uses these techniques would be able to bypass our multi-task consistency detector. However, we noted that these techniques, despite being able to fool both tasks in silo, do not create cross-task consistent adversarial output (e.g., a fake bounding box in OD does not match the location or size of the fake instance segmentation mask).

**Generalization.**   In this paper, we investigated object detection and instance segmentation models, finding the best model pairs to use in a multi-task consistency detector. We are interested in generalizing the approach to any combination of tasks. Especially, we would like to understand if the recommendations (about the model architectures) generalized across tasks.

**Tuple multi-task consistency.**   We propose to extend the detector with more tasks and investigate how the detection rate correlates to the number of tasks. Though, Ghamizi et al. (2022) demonstrated that what matters the most is not the number of tasks or how they correlate, but how much the tasks individually impact the vulnerability of the model. Indeed, the more vulnerable the tasks in the model are, the less likely adding new tasks increases the robustness of the model; and adding a vulnerable task may actually decrease the robustness of the whole model. Thus, a comprehensive analysis is required to answer this challenge.

## 6 RELATED WORK

In recent years, many defenses were created to detect (Hendrycks & Gimpel, 2016; Liu et al., 2019; Tian et al., 2021; Sperl et al., 2020) or to improve the robustness of vision systems against adversarial perturbations (Hendrycks et al., 2019; Mądry et al., 2018). Especially, a strong emphasis has been put on the security of image classification task. Examples of defenses used in image classification include: use of additional detection networks (Liu et al., 2019), analysis of network output (Hendrycks & Gimpel, 2016; Tian et al., 2021), or use of certain activation patterns within the hidden layers (Sperl et al., 2020). These detection methods focus on the output structure or network topology of an image classifier and are thereby not transferable to more complex vision tasks.

As described earlier, multi-task learning (MTL) (Kendall et al., 2018) tackles a wide range of vision tasks in an efficient way. Mao et al. (2020) showed that MTL increases the adversarial robustness due to the increased difficulty of successfully attacking several tasks. Thus, subsequent work explored other task combinations (Xie et al., 2017; Klingner et al., 2020; Wang et al., 2020; Kumar et al., 2021), or compared the effectiveness of adding different auxiliary tasks (Ghamizi et al., 2022; Gurulingan et al., 2021; Haleta et al., 2021). While the positive effects of MTL on adversarial robustness are quite well-explored, we are the first to check the consistency between outputs from object detection and instance segmentation models, deriving recommendations to select best model pairs.

## 7 CONCLUSION

Vision models are paramount to many applications such as autonomous driving. Their robustness have been shown to be brittle under adversarial setting. From the observation that adversarial inputs yield different effects when fed to different models, we propose an adversarial perturbation detection method based on multi-task perception. We showed an example of our lightweight defense using instance segmentation and object detection tasks. We generated adversarial BDD100k datasets and demonstrated our consistency score can effectively detect perturbations. Then, we empirically identified the optimal model pairs, demonstrating that even if sharing the same backbone, the attack can be detected because of uncoordinated perturbations on both models. The optimal models pair had a 99.9% detection rate. Future work will focus on joint multi-task perturbations and assess the effectiveness of our defense against stronger attacker models.

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

# A  APPENDIX

## A.1  ALGORITHM

Algorithm 1 describes the different steps involved in the computation of the consistency score. The function *IoU* function is the function *box_iou* defined in the *Torchvision* library.

---

**Algorithm 1** Consistency Score Calculation

---

**Input:** $\mathcal{S}_{det}$ //A set of pairs of bounding boxes and labels from object detection
**Input:** $\mathcal{S}_{seg}$ //A set of pairs of bounding boxes and labels from instance segmentation
**Output:** Consistency Score ($C$)
  $|CD| = 0$ // Number of pairs (box and mask)
  $IoU = calc\_iou(\mathcal{S}_{det}, \mathcal{S}_{seg})$ // IoU score and label similarity for between pairs of $\mathcal{S}_{det}$ and $\mathcal{S}_{seg}$
  $IoU = prune(IoU, threshold)$ // Prune each box with all IoU scores below threshold
  $n\_box_{seg}, n\_box_{det} = get\_number(IoU)$ // Get remaining number of boxes for each task
  $|CD| = compute\_n\_pairs(n\_box_{seg}, n\_box_{det})$ // Get total number of pairs
  $C = compute\_c(|CD|, len(\mathcal{S}_{det}), len(\mathcal{S}_{seg}))$

---

## A.2  DATASET: BDD100K

The BDD100K dataset is a public dataset of driving scenes, which contains 100k frames and annotations for 10 vision tasks. Compared with other driving datasets, the BDD100k dataset has a diversity of geography, environment, and weather. Therefore, we use the BDD100k as the benchmark dataset to train the models and evaluate our detection. In particular, we use the 100k subfolder for object detection task, which is split to 70k training, 10k validation and 20k testing images. We also use the 10k subfolder for instance segmentation task, which is split to 7k training, 1k validation and 2k testing images.

## A.3  ADVERSARIAL TRANSFERABILITY

This section presents the distinct impact of the attack on OD and SEG models across all model pairs, focusing on the number and size of the detected objects. As previously mentioned, the attack exhibits high transferability between models with similar architectures and backbones, but it also leads to significant inconsistencies in the model outputs. For instance, Figure 8 shows the attack transfer between FRCNN R50 and MRCNN R50, but the number and size of hallucinated objects across the categories vary. For model pairs with different baseline architectures or backbones, such as FRCNN R50 and MASK2FORMER SwinT in Figure 10, the adversarial dataset optimized on MASK2FORMER does not fool FRCNN R50, whose outputs remain close to the ground truth in terms of number and size. Similarly, the adversarial dataset optimized on the FRCNN model does not fool MASK2FORMER whose object areas are close to ground truth. Although the number of objects is very large, this is due to the poor performance of MASK2FORMER, which predicts a large number of objects even on clean inputs, as shown in Figure 7.

This observation also supports the conclusion in Section 4.2.2. The consistency scores for model pairs with similar architectures are low because the attack transfers between them, resulting in distinct impacts, and thus, high inconsistency. For model pairs with different architectures, the attack does not transfer well, leading to low inconsistency. However, when one model in these pairs performs very poorly even on a clean dataset, it will output many hallucinated objects despite the attack not transferring to it, still resulting in high inconsistency, as seen with FRCNN R50 and MASK2FORMER SwinT.

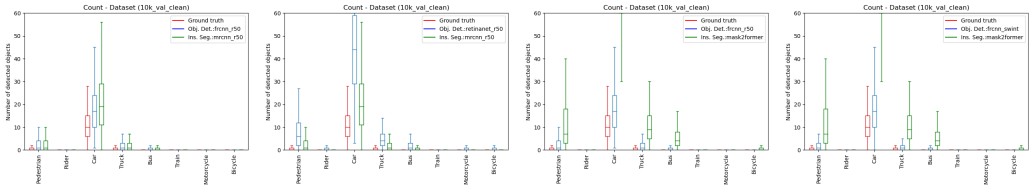

Figure 7: Clean images on model pairs

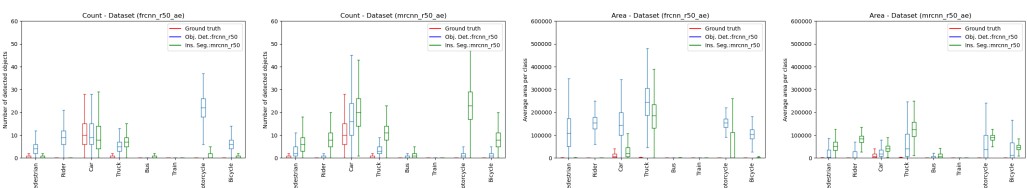

Figure 8: OD_frcnn_r50_SEG_mrcnn_r50

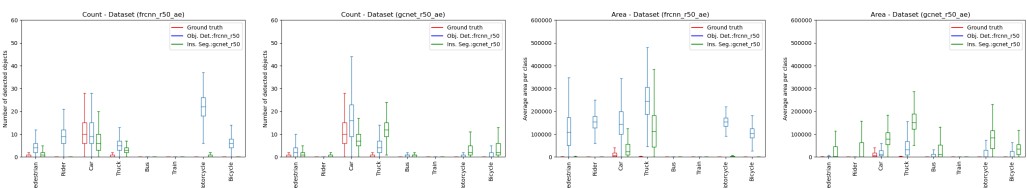

Figure 9: OD_frcnn_r50_SEG_gcnet_r50

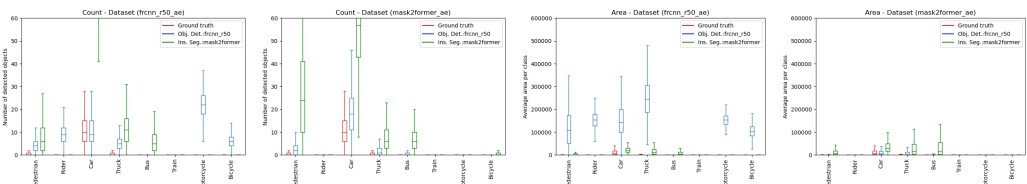

Figure 10: OD_frcnn_r50_SEG_mask2former

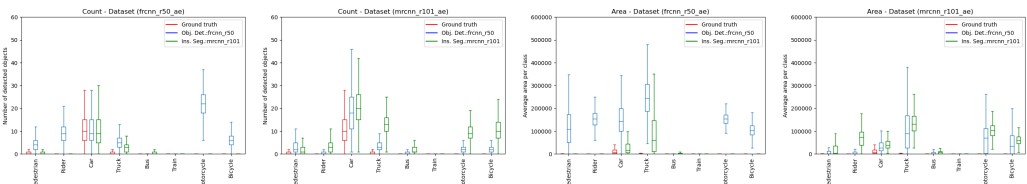

Figure 11: OD_frcnn_r50_SEG_mrcnn_r101

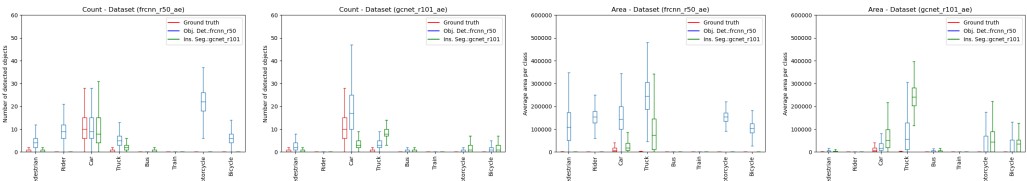

Figure 12: OD_frcnn_r50_SEG_gcnet_r101

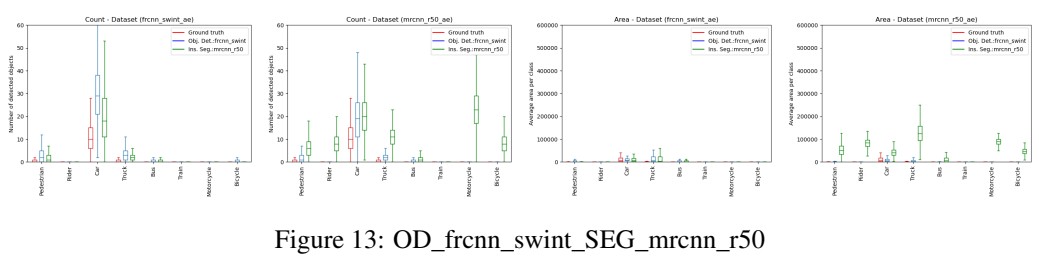

Figure 13: OD_frcnn_swint_SEG_mrcnn_r50

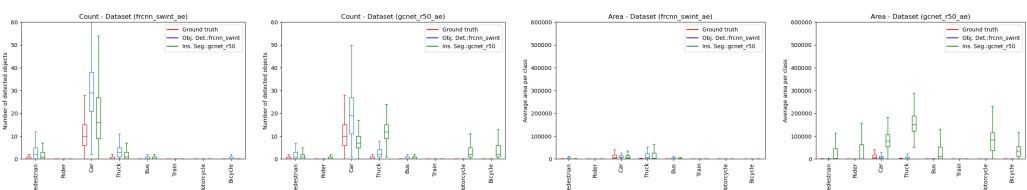

Figure 14: OD_frcnn_swint_SEG_gcnet_r50

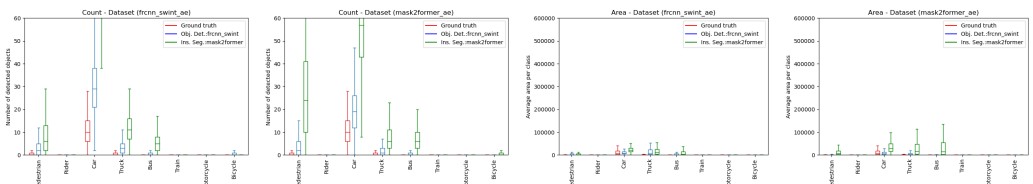

Figure 15: OD_frcnn_swint_SEG_mask2former

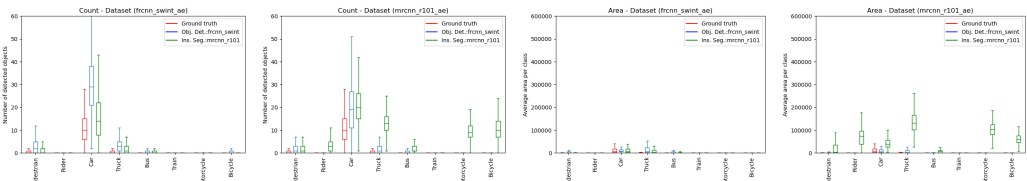

Figure 16: OD_frcnn_swint_SEG_mrcnn_r101

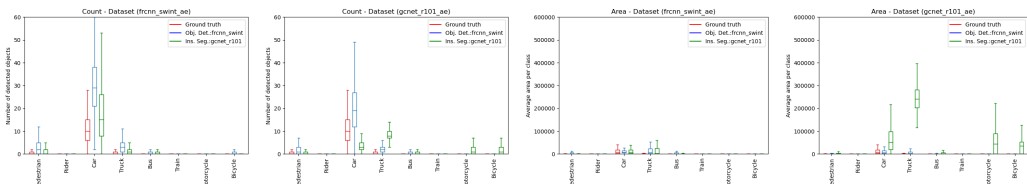

Figure 17: OD_frcnn_swint_SEG_gcnet_r101

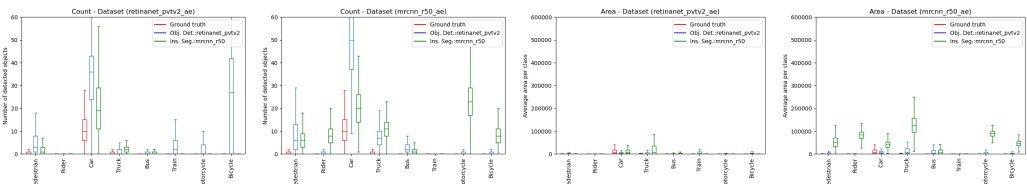

Figure 18: OD_retinanet_pvtv2_SEG_mrcnn_r50

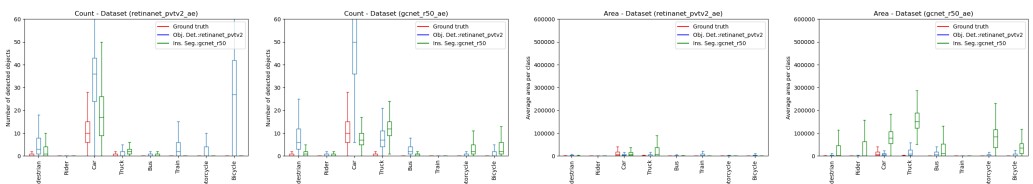

Figure 19: OD_retinanet_pvtv2_SEG_gcnet_r50

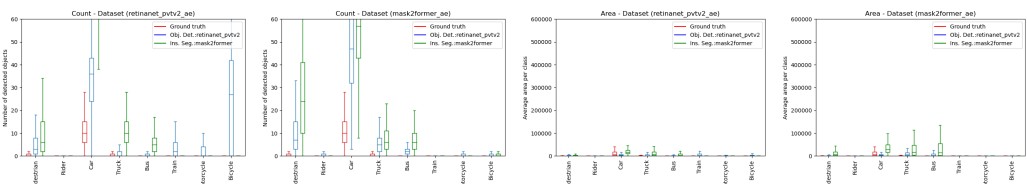

Figure 20: OD_retinanet_pvtv2_SEG_mask2former

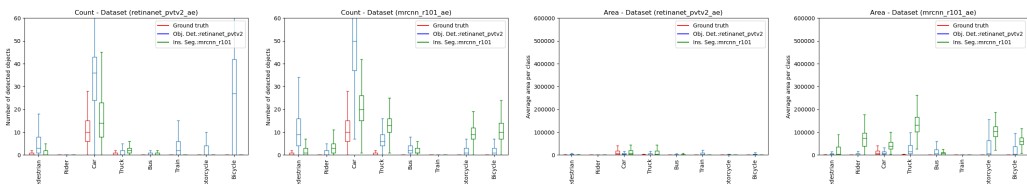

Figure 21: OD_retinanet_pvtv2_SEG_mrcnn_r101

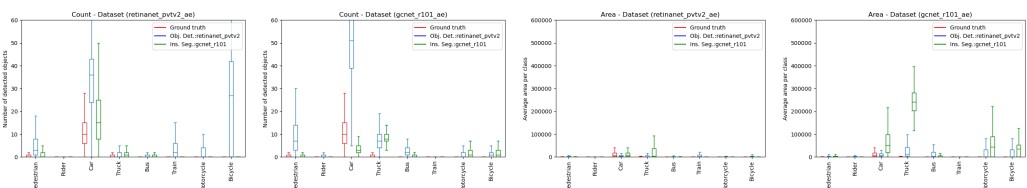

Figure 22: OD_retinanet_pvtv2_SEG_gcnet_r101

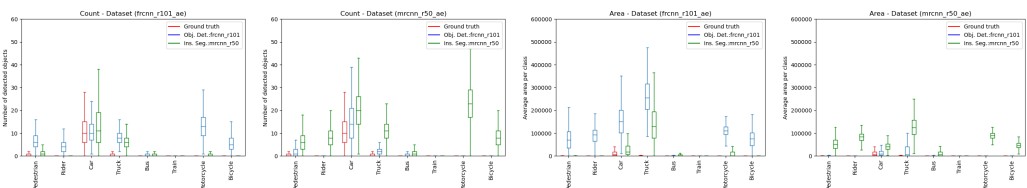

Figure 23: OD_frcnn_r101_SEG_mrcnn_r50

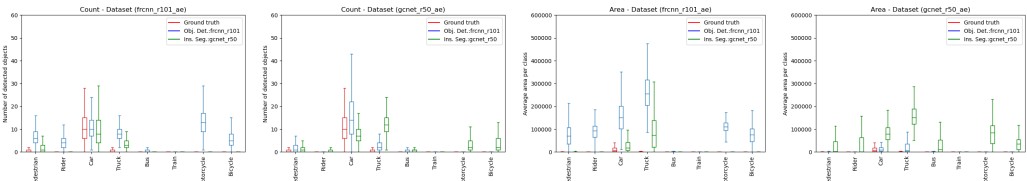

Figure 24: OD_frcnn_r101_SEG_gcnet_r50

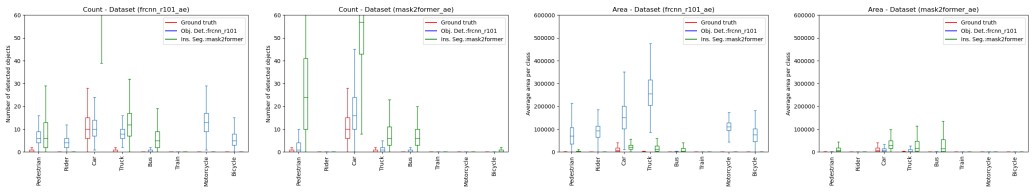

Figure 25: OD_frcnn_r101_SEG_mask2former

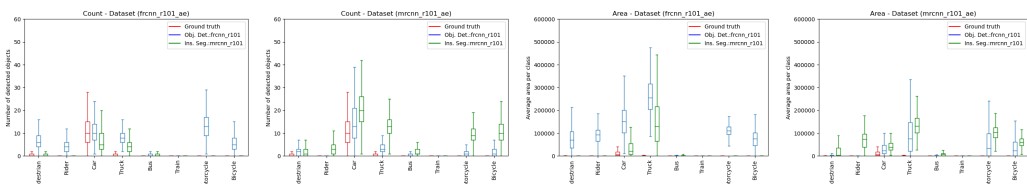

Figure 26: OD_frcnn_r101_SEG_mrcnn_r101

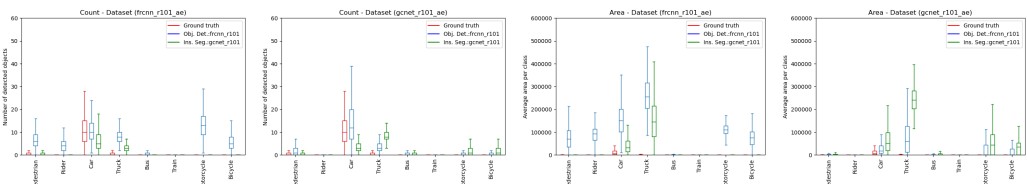

Figure 27: OD_frcnn_r101_SEG_gcnet_r101

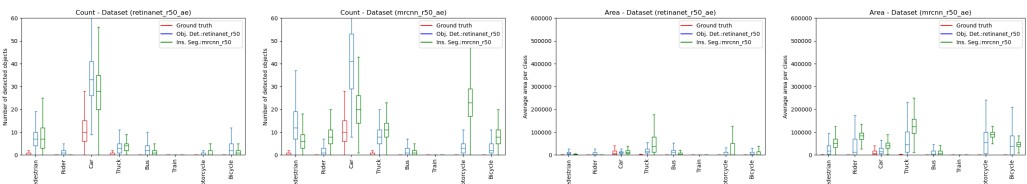

Figure 28: OD_retinanet_r50_SEG_mrcnn_r50

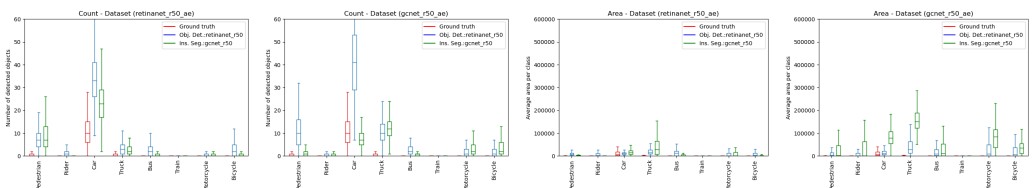

Figure 29: OD_retinanet_r50_SEG_gcnet_r50

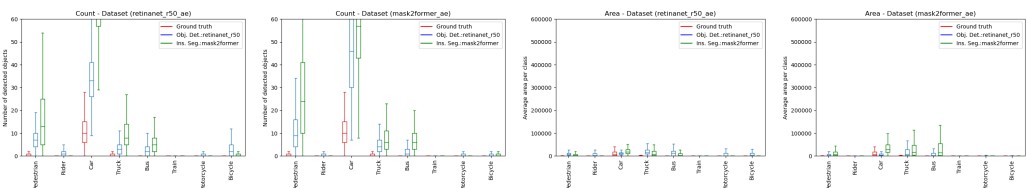

Figure 30: OD_retinanet_r50_SEG_mask2former

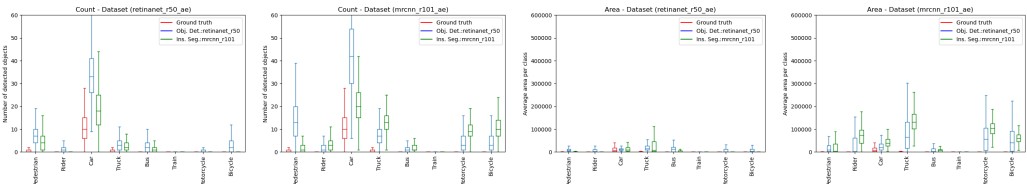

Figure 31: OD_retinanet_r50_SEG_mrcnn_r101

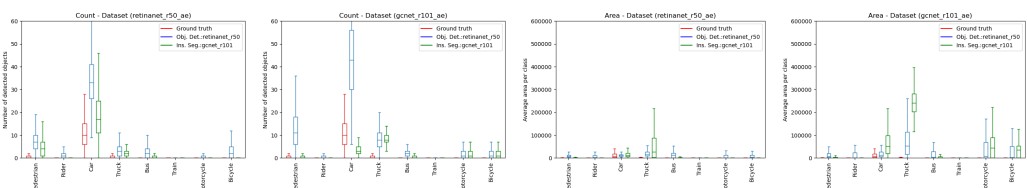

Figure 32: OD_retinanet_r50_SEG_gcnet_r101

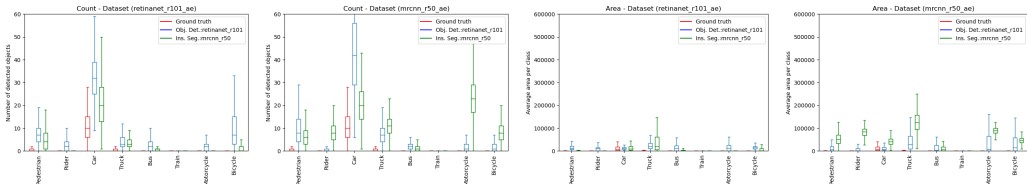

Figure 33: OD_retinanet_r101_SEG_mrcnn_r50

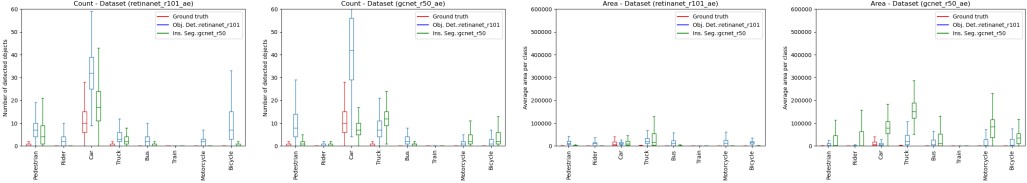

Figure 34: OD_retinanet_r101_SEG_gcnet_r50

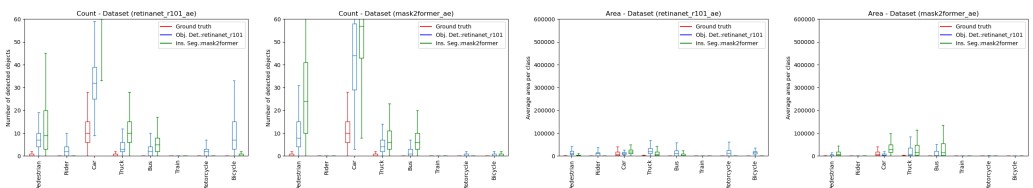

Figure 35: OD_retinanet_r101_SEG_mask2former

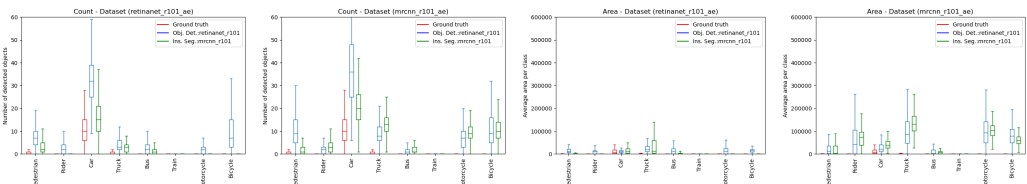

Figure 36: OD_retinanet_r101_SEG_mrcnn_r101

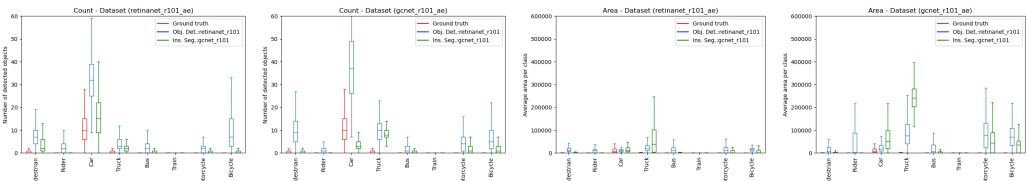

Figure 37: OD_retinanet_r101_SEG_gcnet_r101

## A.4 DETECTOR PERFORMANCE

**Distribution of the consistency score.** This part contains additional results for CS distribution for all model pairs. As previous results showed, the model pairs with similar architecture results in distinct CS distributions between clean inputs and adversarial inputs, e.g., in Figure 38. This is desired for our detector to identify the perturbation. In contrast, the CS distributions for FRCNN R50 and MASK2FORMER SwinT is more difficult to distinguish, particularly when attacking MASK2FORMER SwinT model. This results in a relatively low AUC for this model pair as seen in Figure 5.

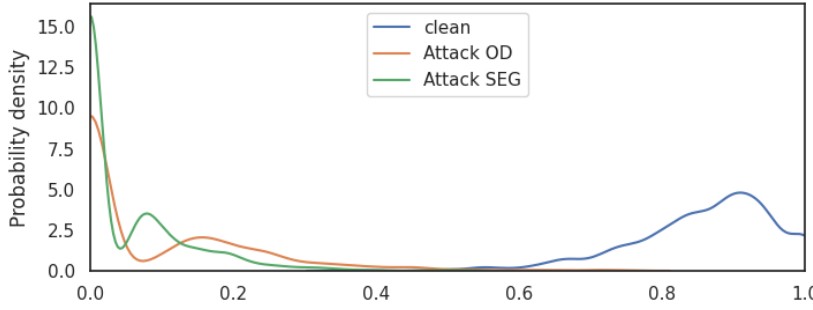

Figure 38: OD_frcnn_r50_SEG_mrcnn_r50

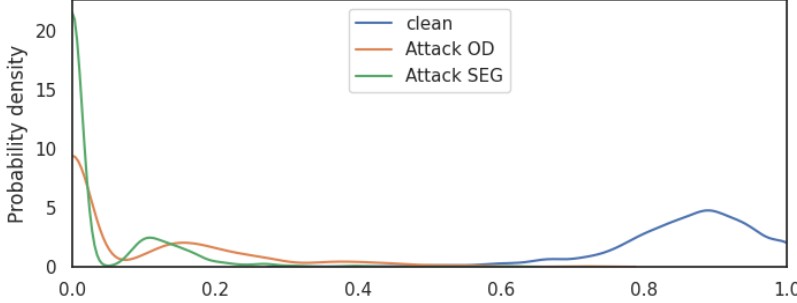

Figure 39: OD_frcnn_r50_SEG_gcnet_r50

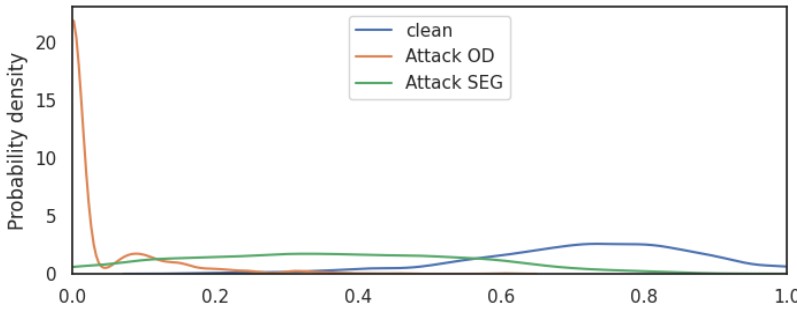

Figure 40: OD_frcnn_r50_SEG_mask2former

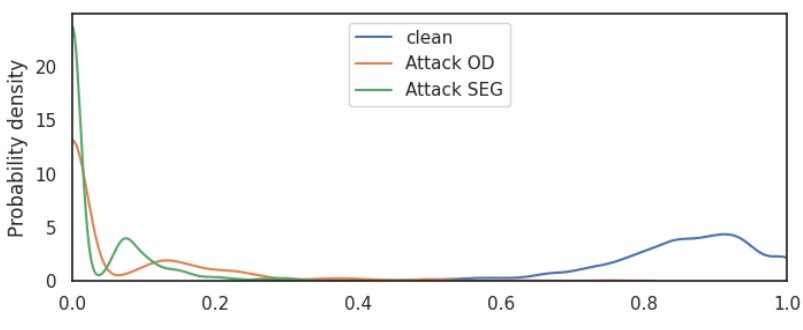

Figure 41: OD_frcnn_r50_SEG_mrcnn_r101

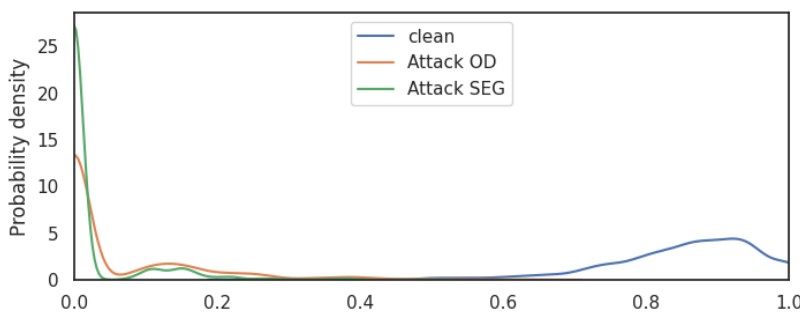

Figure 42: OD_frcnn_r50_SEG_gcnet_r101

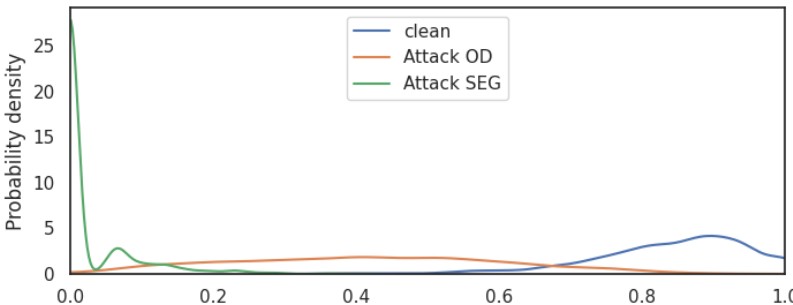

Figure 43: OD_frcnn_swint_SEG_mrcnn_r50

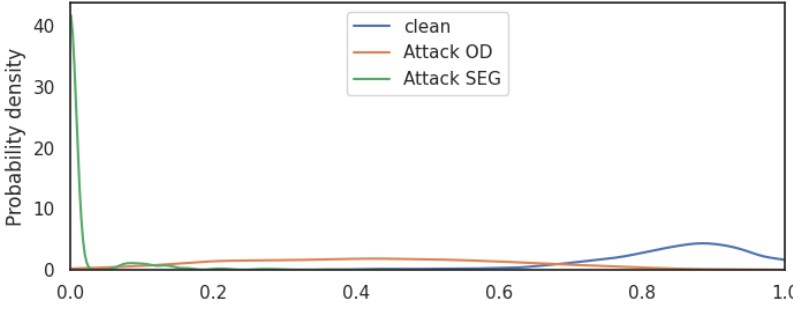

Figure 44: OD_frcnn_swint_SEG_gcnet_r50

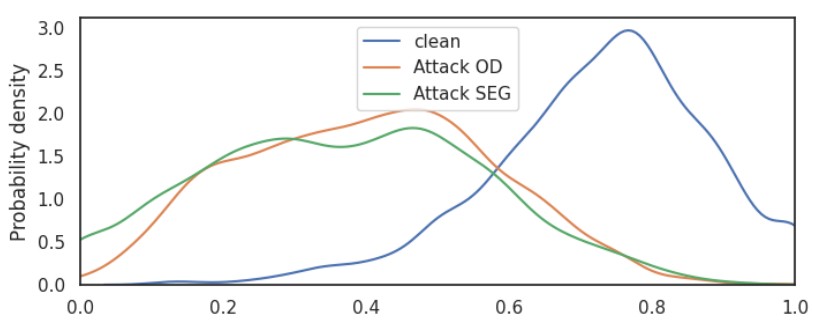

Figure 45: OD_frcnn_swint_SEG_mask2former

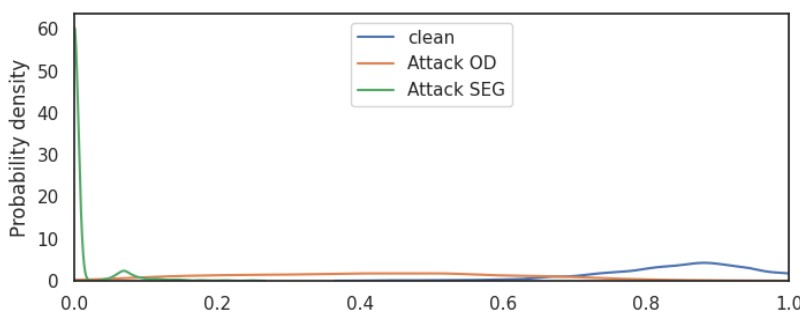

Figure 46: OD_frcnn_swint_SEG_mrcnn_r101

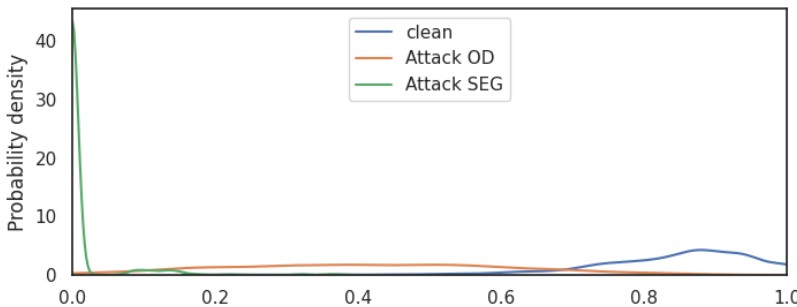

Figure 47: OD_frcnn_swint_SEG_gcnet_r101

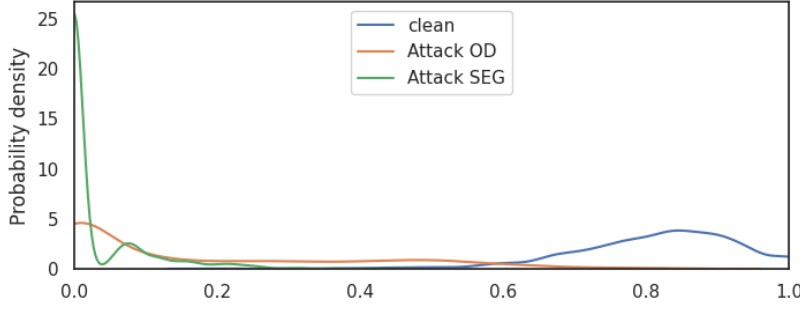

Figure 48: OD_retinanet_pvtv2_SEG_mrcnn_r50

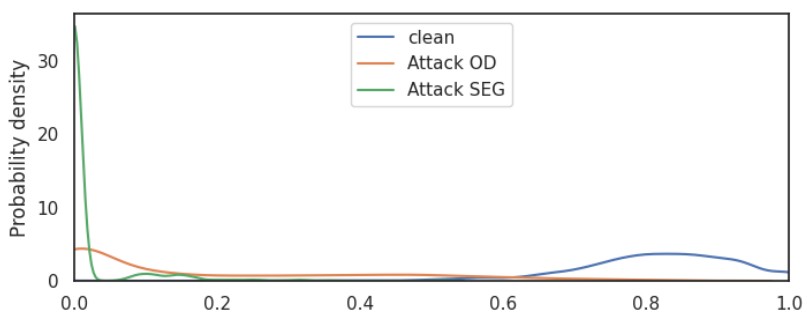

Figure 49: OD_retinanet_pvtv2_SEG_gcnet_r50

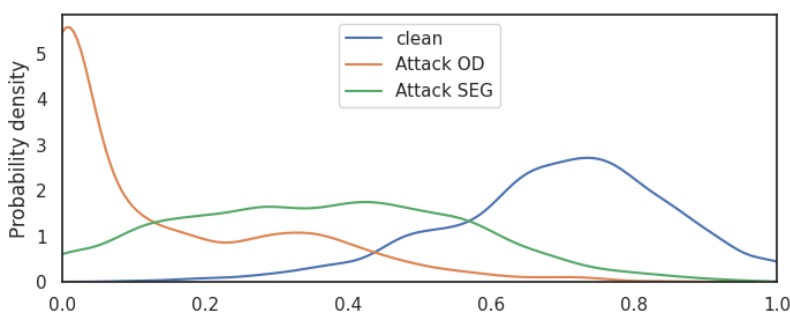

Figure 50: OD_retinanet_pvtv2_SEG_mask2former

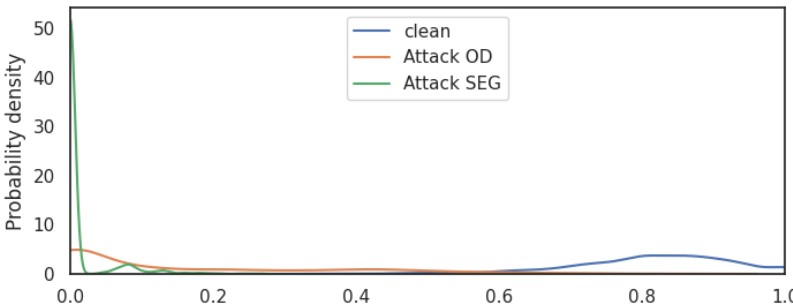

Figure 51: OD_retinanet_pvtv2_SEG_mrcnn_r101

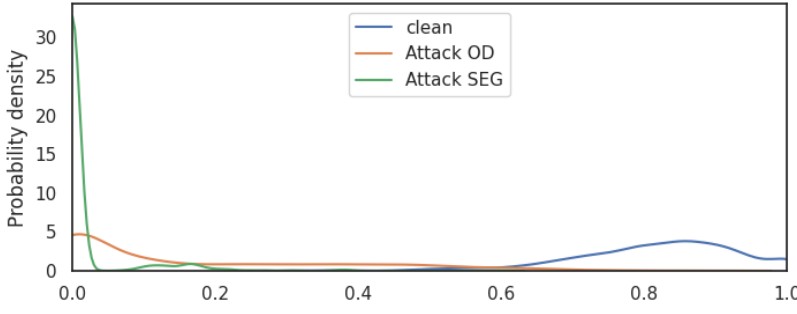

Figure 52: OD_retinanet_pvtv2_SEG_gcnet_r101

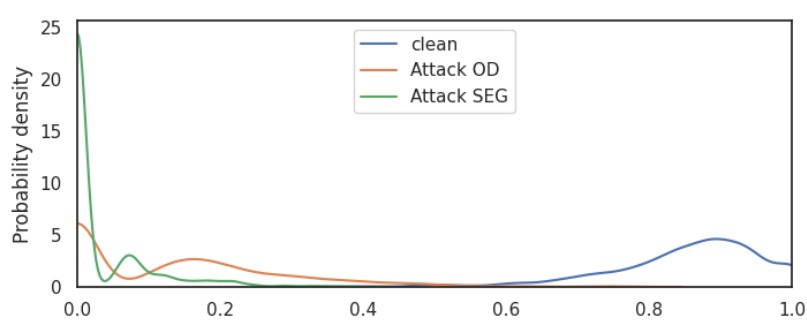

Figure 53: OD_frcnn_r101_SEG_mrcnn_r50

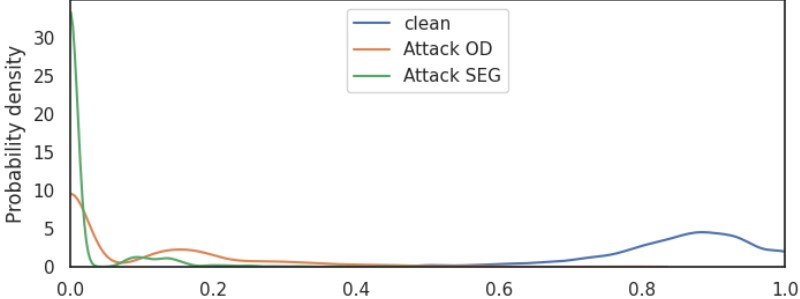

Figure 54: OD_frcnn_r101_SEG_gcnet_r50

**Perturbation strength.** Perturbation strength affects the performance of detector using any model pair. As the perturbation strength increases, it results in stronger impact on the target model and cause higher inconsistency between model pairs.

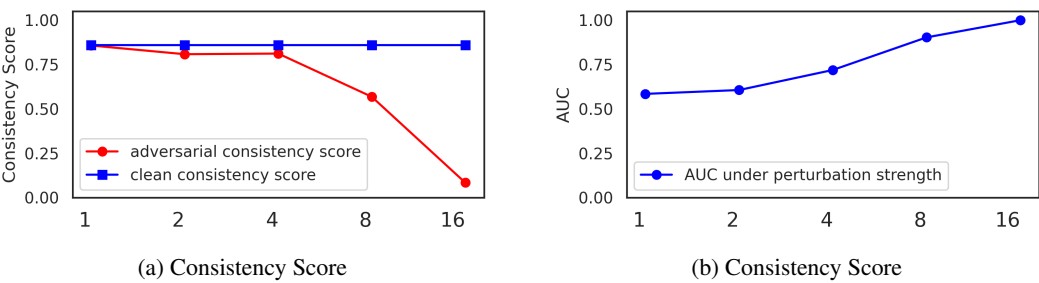

(a) Consistency Score    (b) Consistency Score

Figure 55: Impact of perturbation strength for OD_frcnn_r50_SEG_mrcnn_r50

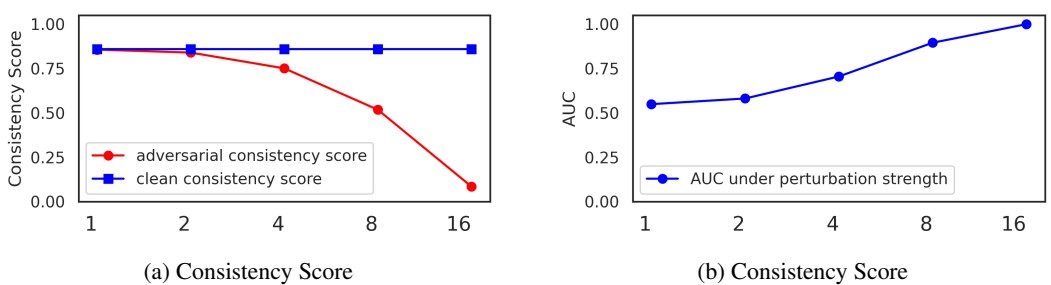

(a) Consistency Score    (b) Consistency Score

Figure 56: Impact of perturbation strength for OD_frcnn_r50_SEG_gcnet_r50

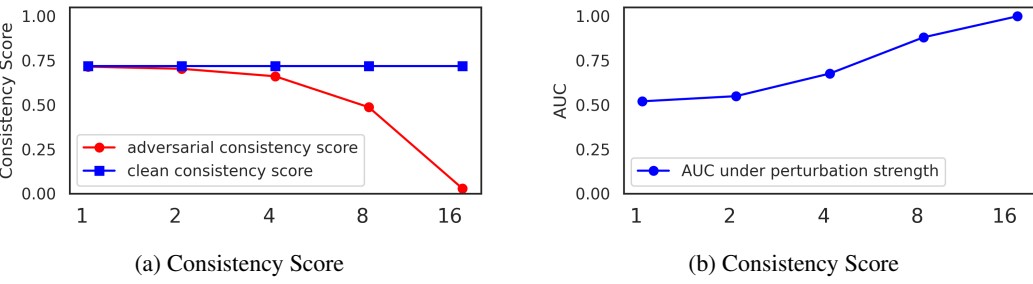

(a) Consistency Score    (b) Consistency Score

Figure 57: Impact of perturbation strength for OD_frcnn_r50_SEG_mask2former

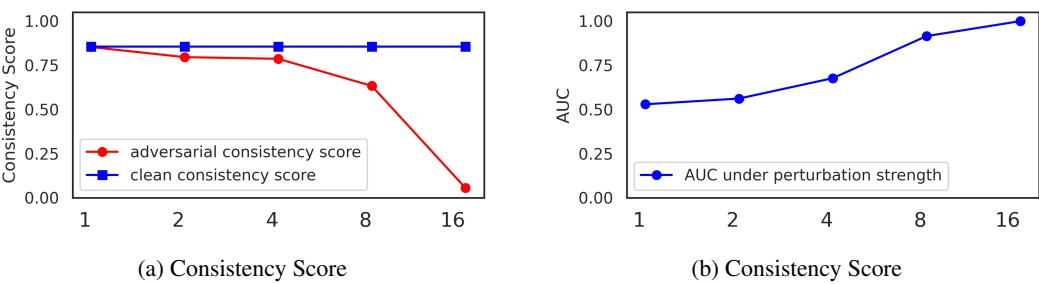

(a) Consistency Score    (b) Consistency Score

Figure 58: Impact of perturbation strength for OD_frcnn_r50_SEG_mrcnn_r101

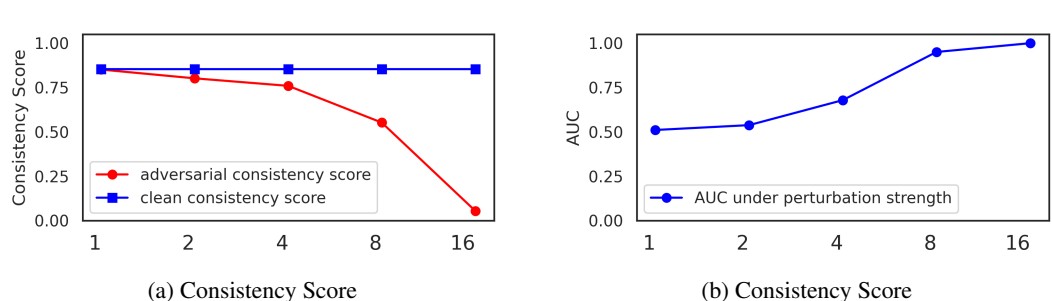

(a) Consistency Score

(b) Consistency Score

Figure 59: Impact of perturbation strength for OD_frcnn_r50_SEG_gcnet_r101

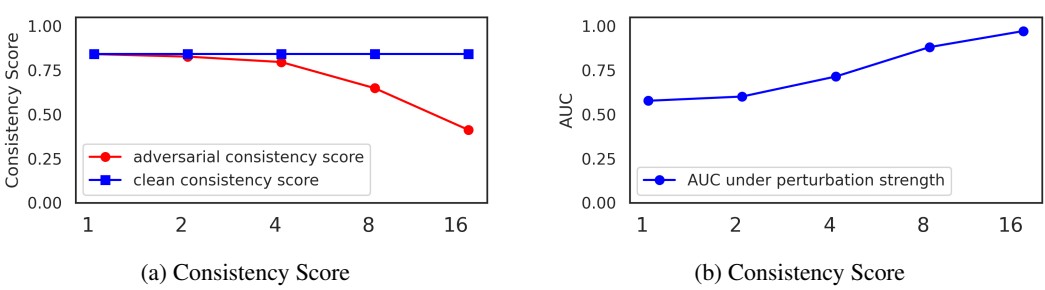

(a) Consistency Score

(b) Consistency Score

Figure 60: Impact of perturbation strength for OD_frcnn_swint_SEG_mrcnn_r50

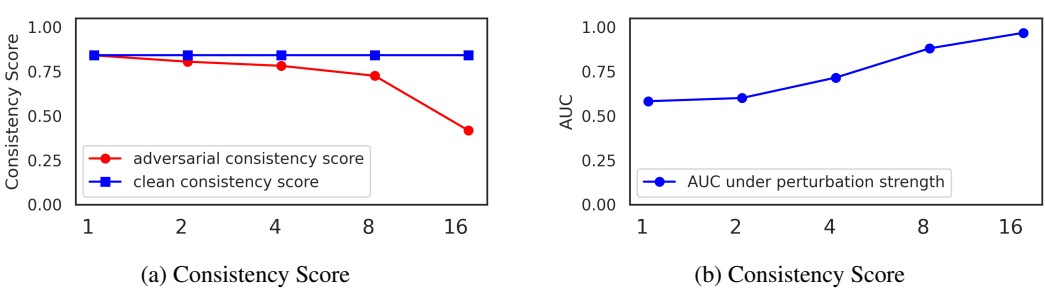

(a) Consistency Score

(b) Consistency Score

Figure 61: Impact of perturbation strength for OD_frcnn_swint_SEG_gcnet_r50

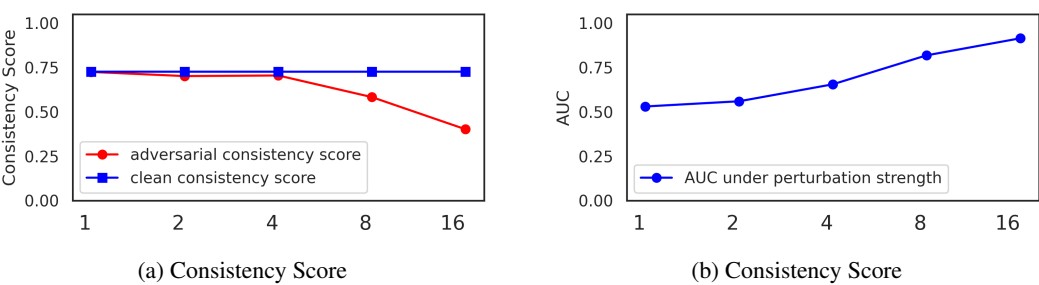

(a) Consistency Score

(b) Consistency Score

Figure 62: Impact of perturbation strength for OD_frcnn_swint_SEG_mask2former

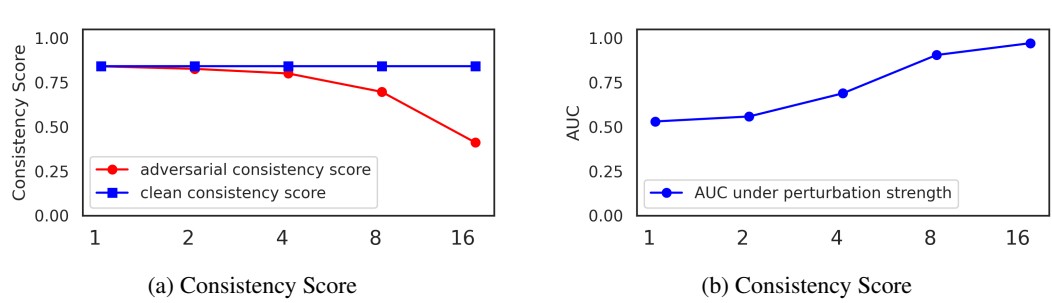

(a) Consistency Score

(b) Consistency Score

Figure 63: Impact of perturbation strength for OD_frcnn_swint_SEG_mrcnn_r101

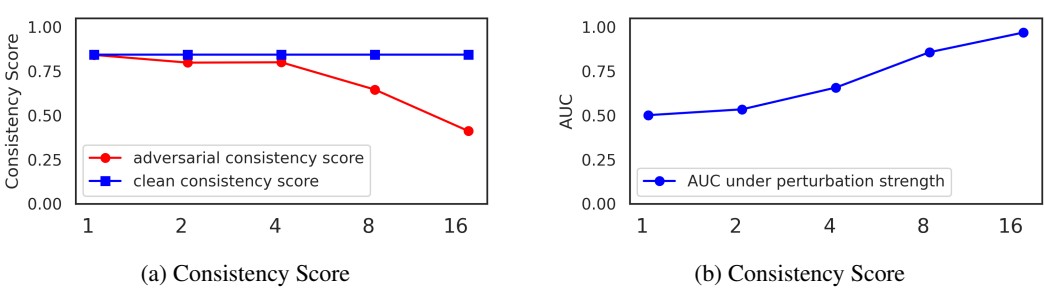

(a) Consistency Score

(b) Consistency Score

Figure 64: Impact of perturbation strength for OD_frcnn_swint_SEG_gcnet_r101

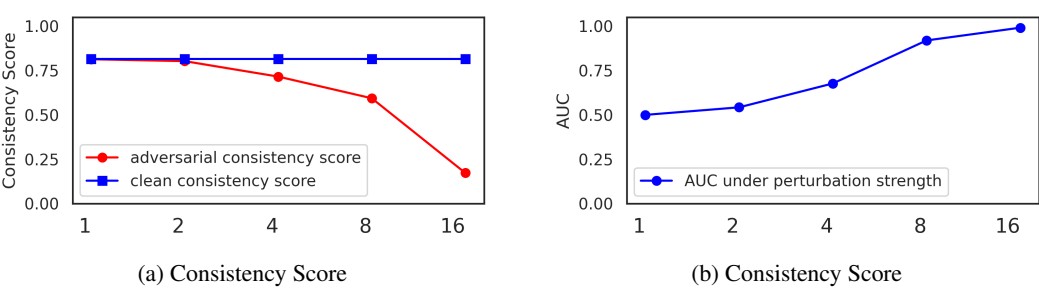

(a) Consistency Score

(b) Consistency Score

Figure 65: Impact of perturbation strength for OD_retinanet_pvtv2_SEG_mrcnn_r50

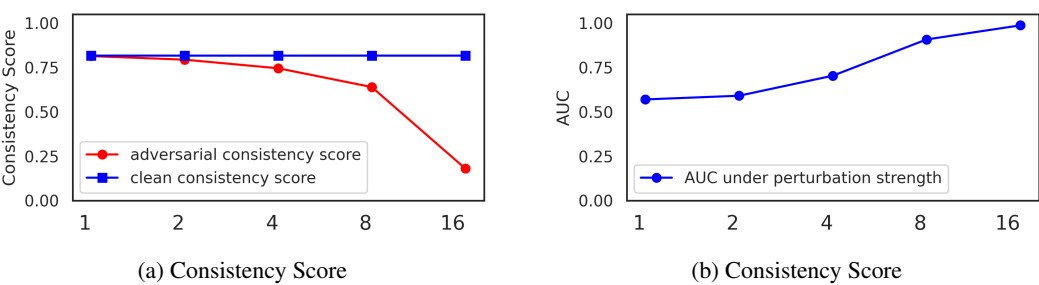

(a) Consistency Score

(b) Consistency Score

Figure 66: Impact of perturbation strength for OD_retinanet_pvtv2_SEG_gcnet_r50

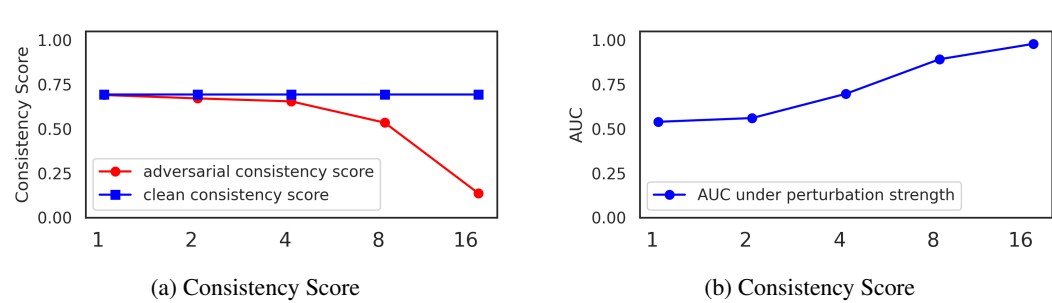

(a) Consistency Score

(b) Consistency Score

Figure 67: Impact of perturbation strength for OD_retinanet_pvtv2_SEG_mask2former

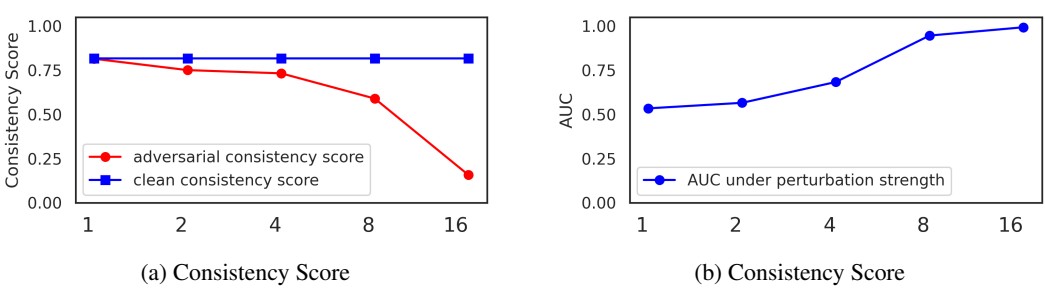

(a) Consistency Score

(b) Consistency Score

Figure 68: Impact of perturbation strength for OD_retinanet_pvtv2_SEG_mrcnn_r101

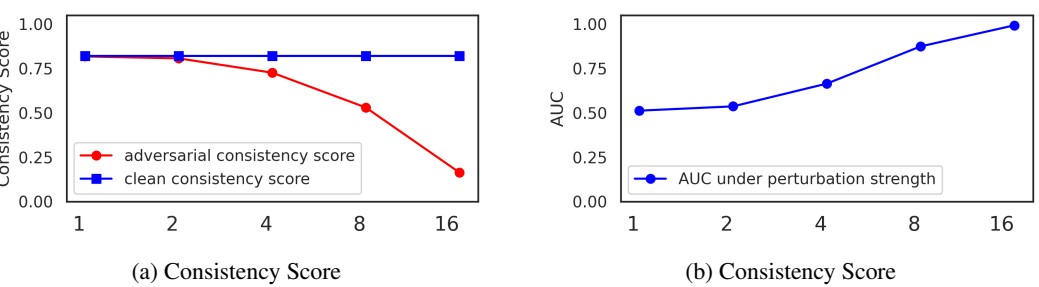

(a) Consistency Score

(b) Consistency Score

Figure 69: Impact of perturbation strength for OD_retinanet_pvtv2_SEG_gcnet_r101

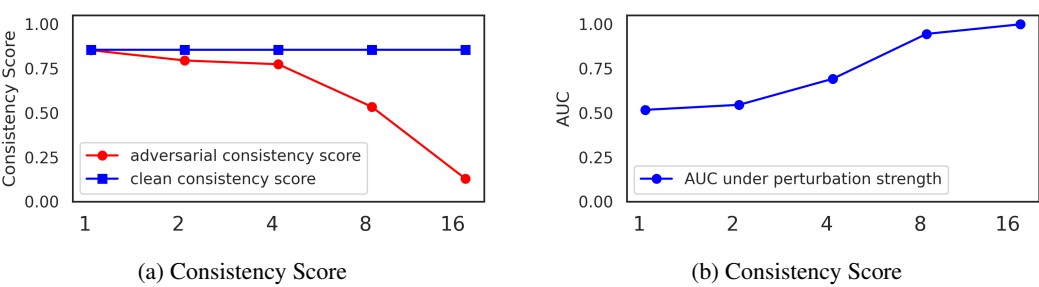

(a) Consistency Score

(b) Consistency Score

Figure 70: Impact of perturbation strength for OD_frcnn_r101_SEG_mrcnn_r50

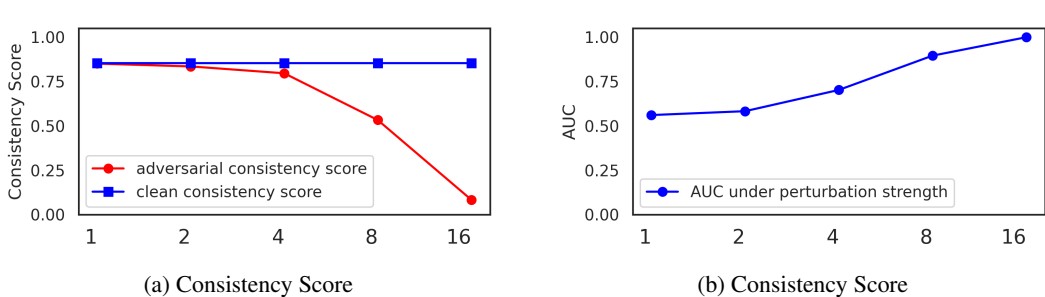

(a) Consistency Score

(b) Consistency Score

Figure 71: Impact of perturbation strength for OD_frcnn_r101_SEG_gcnet_r50

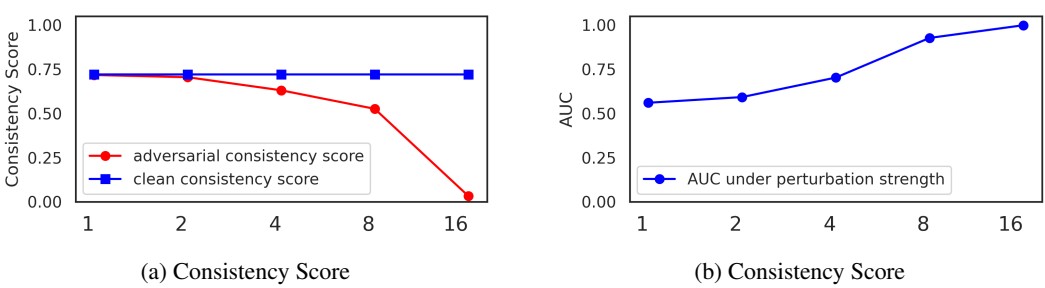

(a) Consistency Score

(b) Consistency Score

Figure 72: Impact of perturbation strength for OD_frcnn_r101_SEG_mask2former

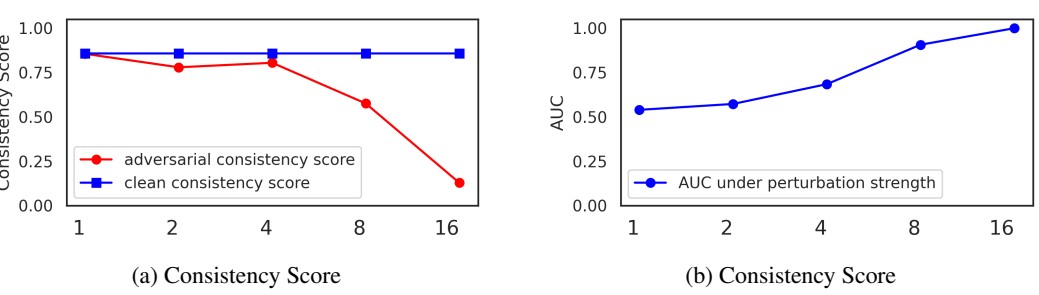

(a) Consistency Score

(b) Consistency Score

Figure 73: Impact of perturbation strength for OD_frcnn_r101_SEG_mrcnn_r101

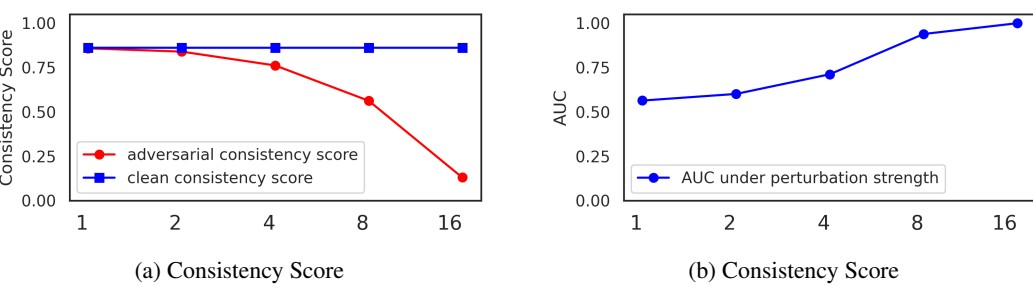

(a) Consistency Score

(b) Consistency Score

Figure 74: Impact of perturbation strength for OD_frcnn_r101_SEG_gcnet_r101

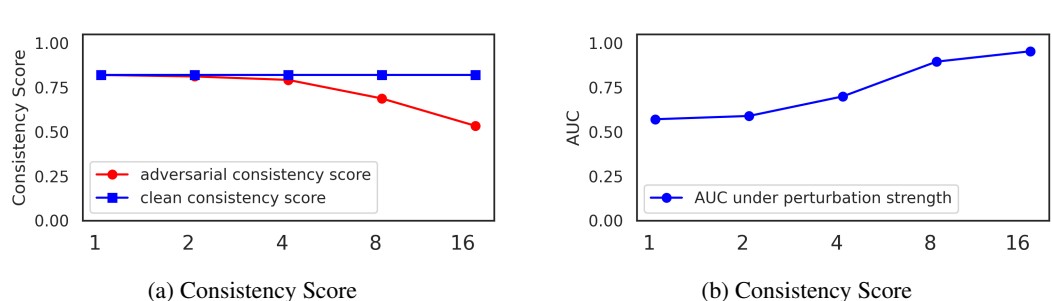

(a) Consistency Score
(b) Consistency Score

Figure 75: Impact of perturbation strength for OD_retinanet_r50_SEG_mrcnn_r50

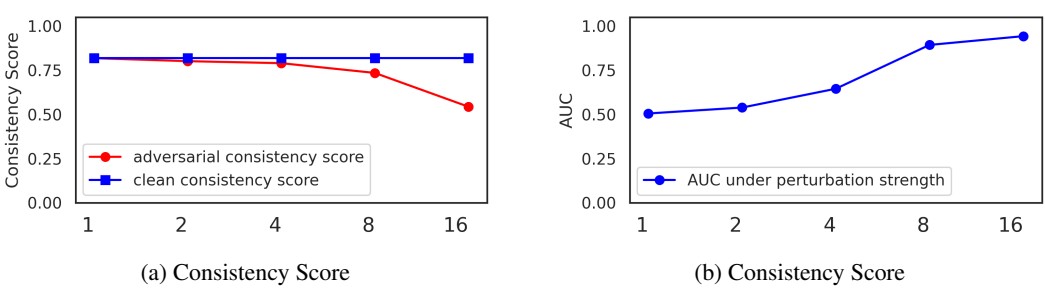

(a) Consistency Score
(b) Consistency Score

Figure 76: Impact of perturbation strength for OD_retinanet_r50_SEG_gcnet_r50

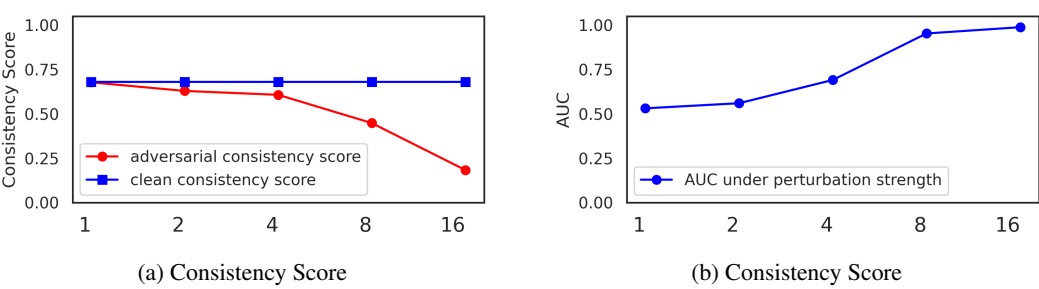

(a) Consistency Score
(b) Consistency Score

Figure 77: Impact of perturbation strength for OD_retinanet_r50_SEG_mask2former

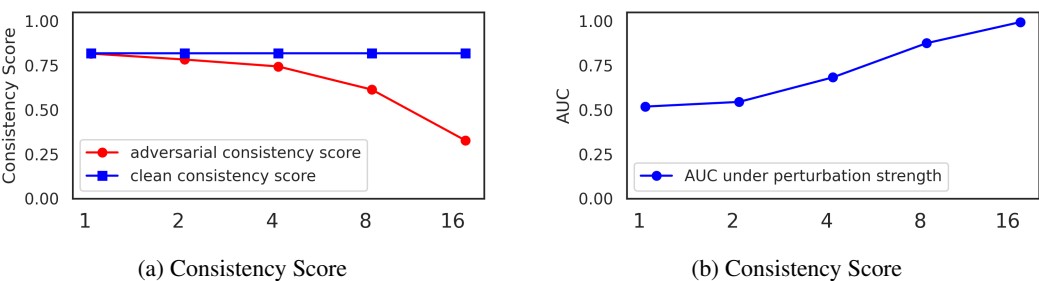

(a) Consistency Score
(b) Consistency Score

Figure 78: Impact of perturbation strength for OD_retinanet_r50_SEG_mrcnn_r101

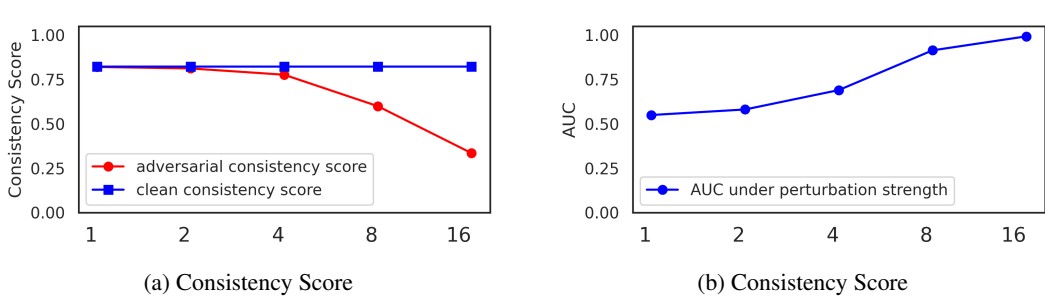

(a) Consistency Score        (b) Consistency Score

Figure 79: Impact of perturbation strength for OD_retinanet_r50_SEG_gcnet_r101

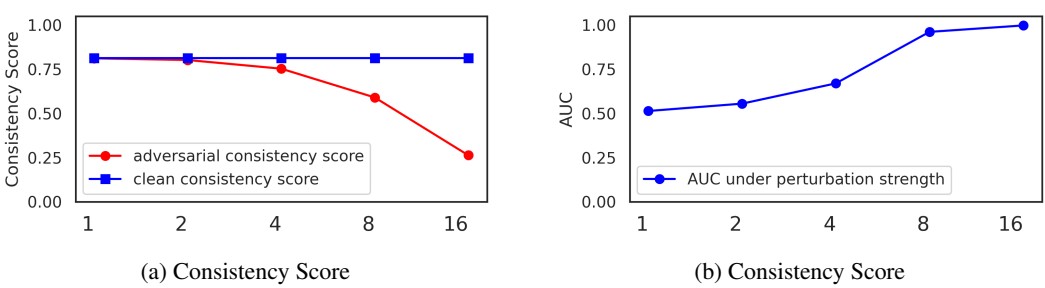

(a) Consistency Score        (b) Consistency Score

Figure 80: Impact of perturbation strength for OD_retinanet_r101_SEG_mrcnn_r50

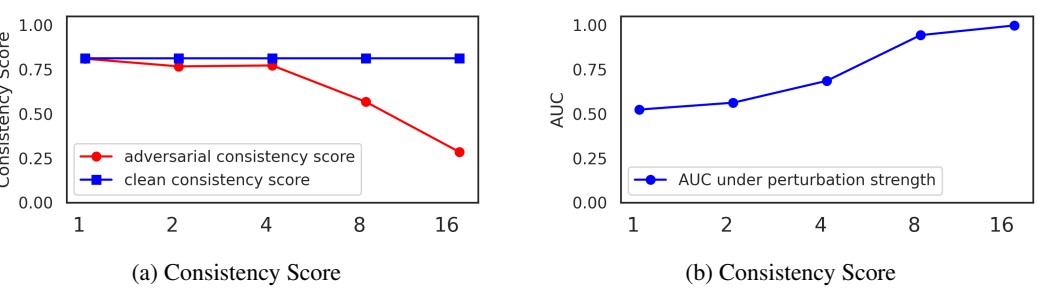

(a) Consistency Score        (b) Consistency Score

Figure 81: Impact of perturbation strength for OD_retinanet_r101_SEG_gcnet_r50

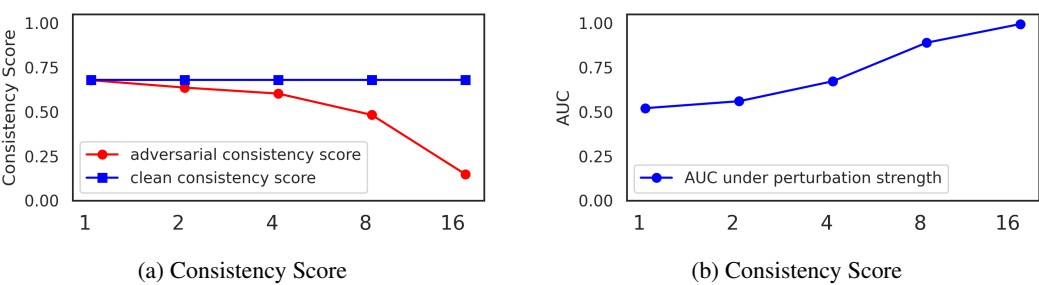

(a) Consistency Score        (b) Consistency Score

Figure 82: Impact of perturbation strength for OD_retinanet_r101_SEG_mask2former

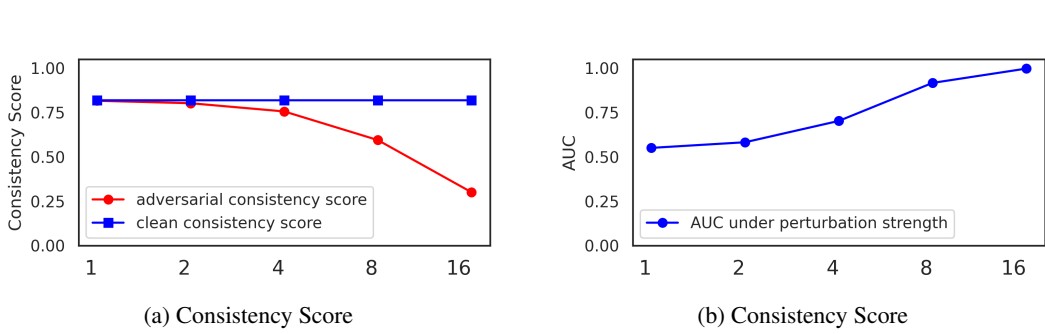

(a) Consistency Score

(b) Consistency Score

Figure 83: Impact of perturbation strength for OD_retinanet_r101_SEG_mrcnn_r101

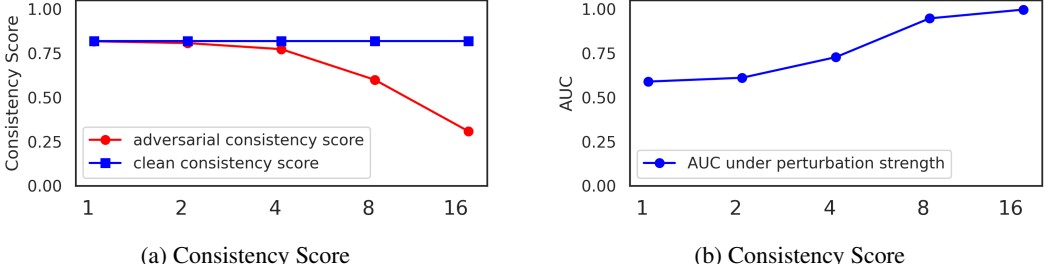

(a) Consistency Score

(b) Consistency Score

Figure 84: Impact of perturbation strength for OD_retinanet_r101_SEG_gcnet_r101

# B  OTHER DEFENSES

To assess the benefits of our multi-task consistency detector, it is important to compare it against other known defenses. Defenses usually fall into three categories: data-based, model-based, detection-based. Data-based defenses use data augmentation at training to improve adversarial robustness. Adversarial training is a common data-based defense (Qian et al., 2022). Model-based focuses on selecting a specific network architecture that provides intrinsic adversarial robustness (Ye et al., 2019; Guo et al., 2020a; Dong et al., 2022). Finally, detection-based defenses do not require data augmentation or model changes, but add a processing (on the input or the output of the model) in order to detect adversarial inputs. PatchCleanser (Xiang et al., 2022) is one example of a double masking technique used to detect presence of adversarial patches in images.

## B.1  ADVERSARIAL TRAINING

The concept of adversarial training (AT) is to train a model on a dataset containing both genuine and adversarial examples in order to build resilience against perturbations. Previous work such as MTD (Zhang & Wang, 2019) and Class-Wise Adversarial Training (CWAT) (Chen et al., 2021) defined loss functions to train the model to accurately localize and classify objects in an image despite the presence of adversarial noise. Unfortunately, all AT schemes demonstrated a drop in model accuracy, which is not desirable.

## B.2  MODEL-BASED DEFENSE: ROBUST NETWORK

### B.2.1  ADVERSARIALLY-AWARE ROBUST OBJECT DETECTOR (ROBUSTDET)

Dong et al. (2022) proposed a counter-proposal to adversarial training by modifying the model architecture. The proposal, named RobustDet, aims to modify an existing backbone (e.g., SSD) by adding three security components: an adversarial image discriminator (AID), an "adversarially-aware convolution" (AAconv), and a consistent features with reconstruction (CFR). The AID is a discriminator that outputs a probability vector based on the category of the image. For instance, if the AID discriminates the image as genuine, then the AID will output the probability vector for a genuine image. Otherwise, if the image is adversarial, then the AID will output the probability vector for an adversarial image. For the training phase, the author formulated a dedicated loss function for the AID to generate a probability vector specific to the category of the image (genuine or adversarial). This probability vector will serve as an input for the next module: *AAconv*. Unlike in adversarial training, *AAconv* aims to use specific weights for the model based on the category of the image. To achieve this goal, *AAconv* uses the concept of dynamic convolution to generate different convolution kernels based on the category of the image. The generation of those convolution kernels is possible thanks to the (genuine or adversarial) probability vector provided by the *AID*. The probability vector serves as the weights to generate convolution kernels. This approach allows to have dedicated weights for genuine images and adversarial images instead of having a single set of weights for both categories of images (like in AT). Lastly, the CFR reconstructs the adversarial image into a clean image.

Looking at their mAP evaluation, RobustDet has higher mAP scores than adversarial training methods such as MTD and CWAT on both genuine and adversarial datasets. However, RobustDet still has at best a 20 mAP score difference between the genuine dataset and the adversarial dataset. This issue means RobustDet do not completely mitigate the mAP loss caused by adversarial examples.

### B.2.2  APPLICATION OF ROBUSTDET ON FASTER RCNN

Dong et al. (2022) evaluated RobustDet on the object detection model SSD with a VGG16 backbone, which was trained on the COCO dataset. However, in our paper, the models evaluated are trained on BDD100k dataset and we do not use SSD. To fairly compare the performance of our consistency-based detector with RobustDet, we applied the *AAconv* technique to one of the models, namely Faster RCNN with RestNet50 backbone (FRCNN R50 in short).

As previously explained, the core idea of *AAconv* is to replace any regular convolution kernel in a model network with a weighted sum of a set of dynamic convolution kernels, expressed as:

$$\dot{\theta}^{AA_{conv}} = \sum_{i=1}^{M} \theta_i^{AA_{conv}} \cdot \pi_i$$

where $\theta_i^{AA_{conv}}$ is the $i$-th kernel in the set of $M$ dynamic kernels, and $\pi_i$ is its corresponding weight generated by *AID*. To clarify, each convolution kernel in the original network will be replaced by a unique set of dynamic kernels whose parameters are determined during the training phase. Thus, for each convolution layer in original Faster RCNN, we replace it with a dynamic convolution layer as defined by Dong et al. (2022). Regarding *AID*, we use the same network architecture Resnet18 as in the original paper. The performance of our Robust FRCNN is presented in Section 4.2.3.

