# OpenReview forum: "Multi-Task Consistency-based Detection of Adversarial Attacks"
_ICLR.cc/2025/Conference — Submitted to ICLR 2025_

### Official Review · Reviewer_vTdF · 2024-10-29

**Soundness:** 3
**Presentation:** 3
**Contribution:** 1
**Rating:** 3
**Confidence:** 5

**Summary:**

This paper introduces an adversarial detection approach utilizing a multi-task model to identify discrepancies between outputs from task-specific heads. The core concept is that adversarial attacks can be detected when the outputs of various visual models are inconsistent. To support this methodology, the authors propose, evaluate, and leverage a new consistency score to quantify output consistency across different vision models. The paper provides results on driving scenarios from the BDD100k dataset, specifically under PGD attacks.

**Strengths:**

- The method, motivation, and novelty of the proposed approach are explained clearly. The introduced consistency score is well-justified with a preliminary study. Additionally, the technique implemented is presented clearly and appears to be reasonable.

- From a technical perspective, applying a multi-task model to driving is a reasonable choice, and designing multi-task consistency metrics to evaluate robustness is a logical approach.

**Weaknesses:**

Overall, I believe the work needs to address significant issues. While I appreciate that the authors have identified some key limitations in the open challenges section, I feel that these should be directly addressed to enhance the paper’s quality. Below are my major concerns:

- The approach relies heavily on using architectures with different backbones. However, the paper lacks discussion or awareness of the computational cost associated with running multiple models with varied backbones (without mixing backbones). This limitation makes it challenging to think about a practical applicability of the method in real-world scenarios as autonomous vehicles.

- The most critical issue is the absence of adaptive or defense-aware attacks. Since the authors propose a defense strategy, it should be tested against a potential worst-case scenario. An adaptive attack could, for example, incorporate the consistency score in an adversarial optimization process to craft specific perturbations aimed at deceiving the multi-task model. For instance see [A].

- Regarding the paper’s contributions, I question the significance of publishing an adversarial dataset based on BDD100k, as it merely involves applying an existing adversarial attack to the dataset.

- Additionally, the methodology is only compared with RobustDet, which was tested on different datasets than those used here. While the BDD dataset does cover more challenging traffic scenarios, including results from more conventional datasets, similar to those used for RobustDet, would provide valuable insights and offer a more comprehensive comparison with the state-of-the-art.

[A] Sobh, Ibrahim, et al. "Adversarial attacks on multi-task visual perception for autonomous driving." arXiv preprint arXiv:2107.07449 (2021).

**Questions:**

- Following the weaknesses regarding attacks, what are the authors' thoughts on adaptive attacks? Why were they not tested? Specifically, what about attacks designed to deceive all components? Is it possible to craft an attack that can fool both individual task heads and the consistency score, for example, by generating perturbations that simultaneously shift the bounding box and semantic segmentation?

- Further studies using different types of attacks could provide valuable insights into the robustness of the technique under varying conditions. What about other types of attacks, such as physically realizable attacks in driving scenarios (e.g., adversarial patches)?

---

### Official Review · Reviewer_qjjM · 2024-11-02

**Soundness:** 3
**Presentation:** 3
**Contribution:** 2
**Rating:** 3
**Confidence:** 4

**Summary:**

This paper proposes a defence method against adversarial attacks on 2D object detection and semantic segmentation. They leverage the inconsistency between the output of a segmentation model and an object detection model as an indicator of the adversarial attacks. Their approach is based on the observation that the clean images usually obtain high consistency across different perception tasks while the adversarial images show low consistency. They have evaluated the defence performance of the combinations of different models and demonstrated their high detection accuracy.

**Strengths:**

1. The presentation of the paper is clear and easy to follow.
2. Their method is direct and performs well in the evaluation.

**Weaknesses:**

1. The threat model is too restricted. Authors only consider PGD attacks against object detection or segmentation models. The proposed defense may be easily bypassed by attacking the two tasks simultaneously, and other more advanced attack methods are not considered.
2. Authors claim that their defense can handle local adversarial attacks, not limited to overall perturbations, but they haven't evaluated this in their experiments.
3. The comparison with other defense methods are limited. Authors only compared their method with an adversarial training one, which I think is not for the same threat model. Adversarial training requires training the target model, but the proposed method is an independent approach. Authors may compare their method with other pre-processing methods or attack detection methods.
4. The evaluation pays much attention on the attack performance, which is not the contribution of this work. The evaluation may need to focus more on the defence performance (e.g., defensive performance against various attacks, comparison with prior defense methods, more ablation studies, etc.)

**Questions:**

Please see the weaknesses.

---

### Official Review · Reviewer_pS7y · 2024-11-04

**Soundness:** 2
**Presentation:** 3
**Contribution:** 2
**Rating:** 3
**Confidence:** 5

**Summary:**

The authors propose using a consistency score to detect adversarial examples against object detection or segmentation models.  The consistency score of each task is defined by the ratio of the total number of consistent detection over the total number of detection. The final consistency score is then defined by the harmonic mean of the scores of the two tasks.

**Strengths:**

1.	The writing is good and easy to understand.
2.	The authors discuss the reasons for using different backbones for different tasks and the limitations of this paper.

**Weaknesses:**

1.	As discussed in Section 5, this paper's key issue is insufficient evaluation. The authors chose only a simple attack for evaluation. Stronger attacks are essential. Please see the questions.
2.	Lack of comparison with previous detection/defense methods.
3.	The authors claimed real-time detection by using only two weak detection models. However, the comparison in Table 4 seems inequitable as the proposed method requires querying a stronger model one more time if the user wants to get reliable final detection results.

**Questions:**

1.	This method seems vulnerable. At least two categories of attacks need to be considered. The first one is transferable attacks, as discussed in section 5. The second is different attack goals. What if the attackers only focus on attacking one object in the image? Such a scenario is vital in the perception system, especially automated driving.
2.	Moreover, it is better to evaluate more realistic attacks, such as physical-realizable patch attacks.

---

### Official Review · Reviewer_EwNo · 2024-11-09

**Soundness:** 3
**Presentation:** 3
**Contribution:** 3
**Rating:** 6
**Confidence:** 4

**Summary:**

The authors propose a new defense mechanism against adversarial attacks on DNNs for AVs, which leverages the inherent multi-task nature of vision perception systems in AVs, exploiting the consistency between different tasks (e.g., object detection and instance segmentation) to detect inconsistencies caused by adversarial perturbations.  A consistency score metric is introduced to measure the level of agreement between the outputs of different vision tasks and demonstrate its effectiveness in detecting attacks.  Authors also provide guidelines for selecting optimal model pairs to maximize the detection performance.  Evaluations on the BDD100k dataset show high accuracy in detecting adversarial perturbations.

**Strengths:**

The approach of leveraging the multi-task nature of vision systems in autonomous driving to defend against adversarial attacks appears to be novel, differing from existing methods that focus on individual tasks or specific types of attacks.

The findings have the potential for improving the security of autonomous driving systems. The proposed defense mechanism appears to be practical and efficient for detecting adversarial attacks. There might be opportunities to apply the key idea to other multi-task learning scenarios.

**Weaknesses:**

The authors only evaluated the security properties under PGD attacks.  Other important attacks such as sensor data spoofing, physical adversarial attacks, data poisoning attacks (e.g., backdoor attacks) are not studied.  I would recommend the authors to at least comment on how the proposed solution can defend against these other classes of attacks.
-> Concrete examples include LiDAR spoofing attacks.

The paper assumes a fairly powerful attacker: a white-box attacker model with access to the model's architecture and weights.   This may not always hold in real-world scenarios. Evaluating the effectiveness of the defense mechanism against stronger attacker models, such as black-box attacks or attacks with limited knowledge of the models, is needed to understand the applicability of this work to more realistic threat models.
Related to this: the paper does not explicitly address the possibility of adaptive attacks, where the attacker modifies their strategy to bypass the defense mechanism. The authors is recommended to comment on the resilience of the proposed method against adaptive attacks and to explore potential countermeasures to such smarter attacks.
-> a specific black-box or limited-knowledge attack scenario that would be particularly relevant for autonomous driving systems could be blackbox LiDAR spoofing attack or camera data spoofing attack.

It's also notable that the authors focus on object detection and instance segmentation as example vision tasks: what about other vision tasks, such as depth estimation or lane detection?

-> discuss how they expect their consistency-based approach would apply to depth estimation or lane detection tasks, and what unique challenges might arise when incorporating these tasks into their multi-task consistency framework.

**Questions:**

The biggest concern is this threat model of white-box attack approach: it seems very implausible to have such an attack be practical.  It's also of concern that the solution does not consider adaptive attacks.
Please comment on these issues.

---

### Meta-Review · Area_Chair_7XkH · 2024-12-21

**Metareview:**

This paper proposed a consistency score based method to detect adversarial examples. The strengths of this paper are (1) clear writing, (2) important topic and (3) simple but effective method. However, this paper has a lot of concerns including (1)  lack of evaluation against adaptive attack, (2) no results on physical attacks, (3) lack of comparison with other baselines;  and (4) restricted threat model. Although authors make some clarification during the rebuttal to address the above concern. However, reviewers still concerned about the adaptive attack evaluation and physical attack evaluation. AC discussed these concerns with reviewers.  Reviewers thoughts the concerns outlined by the reviewers are substantial and seem challenging to be solved without fundamentally reworking the paper. AC agreed with reviewers and hope the authors can improve the paper based on the comments.

**Additional Comments On Reviewer Discussion:**

During the discussion, reviewers still concerned about the experiment results and missing baseline/adaptive attacks/ physical attacks. Reviewers did not address these concerns well.

---

### Decision · Program_Chairs · 2025-01-22

Reject